# Interplay between external inputs and recurrent dynamics during movement preparation and execution in a network model of motor cortex

**Ludovica Bachschmid-Romano[1]\*, Nicholas G Hatsopoulos[2,3], Nicolas Brunel[1,4,5,6]\***

[1]Department of Neurobiology, Duke University, Durham, United States; [2]Department of Organismal Biology and Anatomy, University of Chicago, Chicago, United States; [3]Committee on Computational Neuroscience, University of Chicago, Chicago, United States; [4]Department of Physics, Duke University, Durham, United States; [5]Duke Institute for Brain Sciences, Duke University, Durham, United States; [6]Center for Cognitive Neuroscience, Duke University, Durham, United States

**\*For correspondence:**
ludovica.bachschmid.romano@
duke.edu (LB-R);
nb170@phy.duke.edu (NB)

**Competing interest:** The authors declare that no competing interests exist.

**Abstract** The primary motor cortex has been shown to coordinate movement preparation and execution through computations in approximately orthogonal subspaces. The underlying network mechanisms, and the roles played by external and recurrent connectivity, are central open questions that need to be answered to understand the neural substrates of motor control. We develop a recurrent neural network model that recapitulates the temporal evolution of neuronal activity recorded from the primary motor cortex of a macaque monkey during an instructed delayed-reach task. In particular, it reproduces the observed dynamic patterns of covariation between neural activity and the direction of motion. We explore the hypothesis that the observed dynamics emerges from a synaptic connectivity structure that depends on the preferred directions of neurons in both preparatory and movement-related epochs, and we constrain the strength of both synaptic connectivity and external input parameters from data. While the model can reproduce neural activity for multiple combinations of the feedforward and recurrent connections, the solution that requires minimum external inputs is one where the observed patterns of covariance are shaped by external inputs during movement preparation, while they are dominated by strong direction-specific recurrent connectivity during movement execution. Our model also demonstrates that the way in which single-neuron tuning properties change over time can explain the level of orthogonality of preparatory and movement-related subspaces.

## Editor's evaluation

The study develops a recurrent network model of M1 for center-out reaches, starting from a conventional tuning (or representational) perspective. Through recurrent connectivity, the model shows uncorrelated tuning for movement direction during preparation and execution with the dynamic transition between the two states. The continuous attractor model provides an important example of flexible switching between neural representations and is supported by convincing simulations and analysis.

## Introduction

The activity of the primary motor cortex (M1) during movement preparation and execution plays a key role in the control of voluntary limb movement (*Evarts, 1968*; *Whishaw et al., 1993*; *Whishaw, 2000*; *Graziano et al., 2002*; *Harrison et al., 2012*; *Scott, 2012*; *Brown and Teskey, 2014*). Classic

studies of motor preparation were performed in a delayed-reaching task setting, showing that firing rates correlate with task-relevant parameters during the delay period, despite no movement occurring (*Hanes and Schall, 1996*; *Tanji and Evarts, 1976*; *Churchland et al., 2006a*; *Churchland et al., 2006b*; *Messier and Kalaska, 2000*; *Dorris et al., 1997*; *Glimcher and Sparks, 1992*; *Glimcher and Sparks, 1992*; *Wurtz and Goldberg, 1972*; *Darlington et al., 2018*; *Darlington and Lisberger, 2020*). More recent works have shown that preparatory activity is also displayed before non-delayed movements (*Lara et al., 2018*), that it is involved in reach correction (*Ames et al., 2019*), and that when multiple reaches are executed rapidly and continuously, each upcoming reach is prepared by the motor cortical activity while the current reach is in action (*Zimnik and Churchland, 2021*). Preparation and execution of different reaches are thought to be processed simultaneously without interference in the motor cortex through computation along orthogonal dimensions (*Zimnik and Churchland, 2021*). Indeed, the preparatory and movement-related subspaces identified by linear dimensionality reduction methods are almost orthogonal (*Elsayed et al., 2016*) so that simple linear readouts that transform motor cortical activity into movement commands will not produce premature movement during the planning stage (*Kaufman et al., 2014*). However, response patterns in these two epochs are nevertheless linked, as demonstrated by the fact that a linear transformation can explain the flow of activity from the preparatory subspace to the movement subspace (*Elsayed et al., 2016*). How this population-level strategy is implemented at the circuit level is still under investigation (*Kao et al., 2021*, ). A related open question (*Malonis et al., 2021*) is whether inputs from areas upstream to the primary motor cortex (such as from the thalamus and other cortical regions, here referred to as *external inputs*) that have been shown to be necessary to sustain movement generation (*Sauerbrei et al., 2020*) are specific to the type of movement being generated throughout the whole course of the motor action, or if they serve to set the initial conditions for the dynamics of the motor cortical network to evolve as shaped by recurrent connections (*Churchland et al., 2012*; *Shenoy et al., 2013*; *Hennequin et al., 2014*; *Sussillo et al., 2015*; *Kaufman et al., 2016*; *Vyas et al., 2020*).

In this work, we use a network modeling approach to explain the relationship between network connectivity, external inputs, and computations in orthogonal dimensions. Our analysis is based on electrophysiological recordings from M1 of a macaque monkey performing a delayed center-out reaching task. The dynamics of motor cortical neurons during reaching limb movements has been shown to be low-dimensional (*Gallego et al., 2018*). Here, we develop a low-dimensional description of the dynamics using *order parameters* that quantify the covariation between neural activity and the direction of motion (see *Georgopoulos et al., 1986*; *Schwartz et al., 1988*; *Georgopoulos et al., 1989*; *Georgopoulos et al., 1993* but also *Scott and Kalaska, 1997*; *Scott et al., 2001*). Recorded neurons are tuned to the direction of motion both during movement preparation and execution, but their preferred direction and amplitude of the tuning function change over time (*Hatsopoulos et al., 2007*; *Churchland and Shenoy, 2007*; *Rickert et al., 2009*). Interestingly, major changes happen when the activity flows from the preparatory to the movement-related subspaces. We describe neuronal selectivity during the task by four parameters: two angular variables corresponding to the preferred direction during movement preparation and execution, respectively; and two parameters that represent the strength of tuning in the two epochs. We characterized the empirical distribution of these parameters, and investigated potential network mechanisms that can generate the observed tuning properties by building a recurrent neural network model, whose synaptic weights depend on tuning properties of pre and post-synaptic neurons, and external inputs can contain information about movement direction. First, we analytically derived a low-dimensional description of the dynamics in terms of a few observables denoted as *order parameters*, which recapitulate the temporal evolution of the population-level patterns of tuning to the direction of motion. Then, we inferred the strength of recurrent connections and external inputs from data, by imposing that the model reproduce the observed dynamics of the order parameters. There are multiple combinations of feedforward and recurrent connections that allow the model to generate neural activity that strongly resembles the one from recordings both at the single-neuron and population level, and that can be transformed into realistic patterns of muscle activity by a linear readout. To break the model degeneracy, we imposed an extra cost associated with large external inputs – that likely require more metabolic energy consumption compared to local recurrent inputs. The resulting solution suggests that different network mechanisms operate during movement preparation and execution. During the delay period, the population activity is shaped by external inputs that are tuned to the preferred directions of the neurons. During

movement execution, the localized pattern of activity is maintained via strong direction-specific recurrent connections. Finally, we show how the specific way in which neurons tuning properties rearrange over time produces the observed level of orthogonality between the preparatory- and movement-related subspaces.

## Results

### Subjects and task

We analyzed multi-electrode recordings from the primary motor cortex (M1) of two macaque monkeys performing a previously reported instructed-delay, center-out reaching task (*Rubino et al., 2006*). The monkey's arm was on a two-link exoskeletal robotic arm, so that the position of the monkey's hand controlled the location of a cursor projected onto a horizontal screen. The task consisted of three periods (*Figure 1a*): a hold period, during which the monkey was trained to hold the cursor on a center target and wait 500ms for the instruction cue; an instruction period, during which the monkey was presented with one of eight evenly spaced peripheral targets and continued to hold at the center for an additional 1,000–1,500ms; a movement period, signalled by a go cue, when the monkey initiated the reach to the peripheral target. Successful trials for which the monkeys reached the target were rewarded with a juice or water reward. The peripheral target was present on the screen throughout the whole instruction and movement periods. In line with previous studies (e.g. *Elsayed et al., 2016*), the preparatory and movement-related epochs were defined as two 300ms time intervals beginning, respectively, 100ms after target onset and 50ms before the start of the movement.

### Patterns of correlation between neural activity and movement direction

Studies of motor cortex have shown that tuning to movement direction is not a time-invariant property of motor cortical neurons, but rather varies in time throughout the course of the motor action; single-neuron encoding of entire movement trajectories has been also reported (e.g. [*Hatsopoulos et al., 2007*; *Churchland and Shenoy, 2007*]). We measured temporal variations in neurons preferred direction by binning time into $160ms$ time bins and fitting the binned trial-averaged spike counts as a function of movement direction with a cosine function. As the variability in preferred direction during the delay period alone and during movement execution alone was significantly smaller than the variability during the entire duration of the task (*Figure 1—figure supplement 1*), we characterized neurons tuning properties only in terms of these two epochs. This simplifying assumption is discussed further in Methods and Discussion. We will show later that such a simplification is enough for the model to recapitulate neuronal activity both at the level of single-units and at the population level.

*Figure 1b–c* show two examples of tuning curves, where the trial-averaged and time-averaged activity during movement preparation and execution is plotted as a function of the location of the target on the screen, together with a cosine fitting curve. In the two epochs, neurons change their preferred direction - denoted, respectively, as $\theta_A$ and $\theta_B$ - and their degree of participation, which is proportional to the amplitude of the cosine tuning function - denoted as $\eta_A$ and $\eta_B$. A scatter plot of the preferred directions in preparatory ($\theta_A$) and execution ($\theta_B$) epochs for all neurons is shown in *Figure 1d*, while a similar scatter plot for the degrees of tuning in preparatory ($\eta_A$) and execution ($\eta_B$) epochs is shown in *Figure 1e*. Neurons preferred direction in the two epochs are moderately correlated (circular correlation coefficient $r = 0.4$, $p = 3\,10^{-5}$), and so are their degree of participation, even if to a lesser degree (correlation coefficient $r = 0.3, p = 10^{-4}$). Neuronal tuning to movement direction is reflected in a population activity profile that is spatially localized, as shown in *Figure 1f-g*, were we plot the normalized firing rate for all neurons, as a function of their preferred direction, separately for the two epochs. An example of the activity of a single neuron in the two epochs is highlighted in red. It illustrates how the activity of single neurons is drastically rearranged across time, while the population activity remains localized around the same angular location, which corresponds to the location of the target on the screen.

### Recurrent neural network model

The experimental observations described above motivated us to build a network model in which neuronal selectivity properties match the ones observed in the data. In particular, we built a network

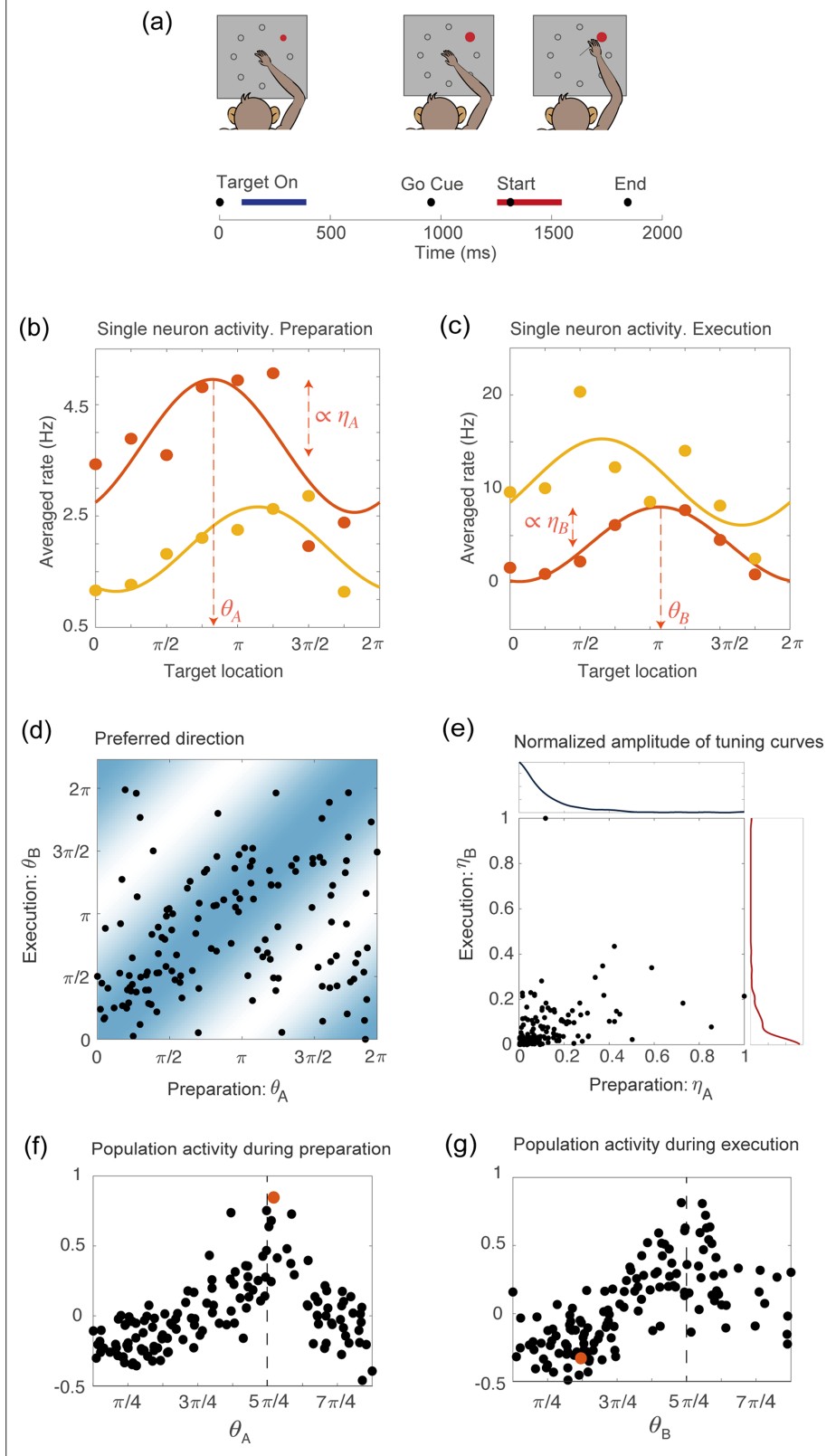

**Figure 1.** Tuning to the direction of motion during movement preparation and execution in a delayed reaching task. (**a**) Schematic of the center-out delayed reaching task and definition of the preparatory (blue) and of the movement-related (red) epochs. Black circles represent the time of: target onset; go cue; start of movement; end of movement, averaged across trials. (**b**) Example of the trial averaged firing rate of two neurons during movement

*Figure 1 continued on next page*

*Figure 1 continued*

preparation as a function of the target location. Solid lines represent the corresponding cosine fit. For one example neuron, we show its preferred direction ($\theta_A$), defined as the location of the peak of the cosine function; and its degree of participation ($\eta_A$), which is proportional to the amplitude of the cosine function. (**c**) Same as in (**a**), but for the activity of the same neurons during movement execution. The preferred direction is denoted as $\theta_B$ and the degree of participation is $\eta_B$. (**d**) Scatter plot of the preferred direction during execution ($\theta_B$) vs preparation ($\theta_A$) for all neurons; preferred directions are defined as the location of the peak of the cosine tuning functions (circular correlation coefficient $r = 0.4$). The heat map represents the joint distribution over ($\theta_A, \theta_B$) inferred from the data (*Equation 4*). (**e**) Scatter plot of the normalized amplitude of the cosine tuning curve during execution ($\eta_B$) vs preparation ($\eta_A$) for all neurons (correlation coefficient $r = 0.3$). Blue line: empirical distribution of $\eta_A$ from kernel density estimation; red line: empirical distribution of $\eta_B$. (**f–g**) Trial averaged firing rate for all neurons during movement preparation as a function of $\theta_A$ (**f**) and $\theta_B$ (**g**), for the condition with target location $= 5\pi/4$, indicated by the dashed vertical line. The activity is normalized across condition for each neuron. The activity of one neuron chosen at random is indicated by the orange dots in panels f and g.

The online version of this article includes the following figure supplement(s) for figure 1:

**Figure supplement 1.** Variance of preferred directions across time.

**Figure supplement 2.** Distribution of tuning parameters from data.

model in which neurons are characterized by the four same parameters we use to fit the preferred directions $\theta_A, \theta_B$ and degrees of tuning, $\eta_A, \eta_B$, leading to a 4-dimensional selectivity space. The subspace defined by the coordinates ($\theta_A, \eta_A$) is denoted as map $A$, while the subspace defined by ($\theta_B, \eta_B$) is denoted as map $B$. We will denote the coordinate vector by

$$x = \{\theta_A, \theta_B, \eta_A, \eta_B\},$$

and refer to the units in the network as neurons, even though each unit in the model could instead represents a group of M1 neurons with similar functional properties, and likewise the connection between two units in our model could represent the effective connection between the two functionally similar groups of neurons in M1.

In this model, neurons with coordinates (selectivity parameters) $x$ are described by their firing rate $r(x, t)$ whose temporal evolution is given by:

$$\tau \tfrac{d}{dt} r(x; t) = -r(x; t) + \left[ I^{\text{tot}}(x; t) \right]_+ , \tag{1}$$

where $\tau$ is the time constant of firing rate dynamics. $[\ ]_+$ is the threshold-linear (a.k.a. relu) transfer function that converts synaptic inputs in firing rates, and $I^{\text{tot}}(x; t)$ is the total synaptic input to neurons with coordinates $x$. We set the time constant to $\tau = 25$ ms, which is of the same order of magnitude of the membrane time constant, and we checked that for values of $\tau$ in the range $10ms - 100ms$ our results did not quantitatively change. The total input to a neuron is

$$I^{\text{tot}}(x; t) = \int dx' \rho(x') J(x, x') r(x'; t) + I^{\text{ext}}(x; t). \tag{2}$$

The first term in the r.h.s. of (2) is the recurrent input, which depends on the firing rates of presynaptic neurons $r(x', t)$, and on $J(x, x')$, the strength of recurrent connections from neurons with coordinates $x'$ to neurons with coordinates $x$. $I^{\text{ext}}(x; t)$ is the external input, and $\rho(x)$ is the probability density of $x$.

The recurrent term in $I^{\text{tot}}$ is the sum of single neuron contributions: here, we assumed that the cortical network is sufficiently large that instead of summing over the contributions of single neurons, each one at a given coordinate $x = \{\theta_A, \theta_B, \eta_A, \eta_B\}$, we can integrate over a continuous distribution of $x$. The probability density of the coordinates $\rho(x)$ is set to match the empirical distribution of the preferred direction and degree of participation. The preferred directions are not significantly correlated with the degrees of tuning (*Figure 1—figure supplement 2*), and we therefore took them to be independent:

$$\rho(x) = \rho_d(\theta_A, \theta_B)\, \rho_p(\eta_A, \eta_B). \tag{3}$$

The distribution of preferred directions was well fitted by:

$$\rho_d(\theta_A, \theta_B) = \frac{1}{4\pi^2}\left[1 + \frac{2}{3}\cos(\theta_A - \theta_B)\right],\tag{4}$$

as shown in *Figure 1—figure supplement 2*, while for the sake of simplicity we assumed $\rho_p(\eta_A, \eta_B) = \rho_{pA}(\eta_A)\,\rho_{pB}(\eta_B)$ and estimated $\rho_{pA}, \rho_{pB}$ non-parametrically using kernel density estimation. The last two ingredients that we need to specify to define the network dynamics are the recurrent couplings and external inputs. The strength of synaptic connections from a pre-synaptic neuron with preferred directions $(\theta'_A, \theta'_B)$ and participation strengths $(\eta'_A, \eta'_B)$ to a post-synaptic neuron with preferred directions $(\theta_A, \theta_B)$ and participation strengths $(\eta_A, \eta_B)$ is given by

$$J(x', x) = j_0 + j_s^A \eta_A \eta'_A \cos(\theta_A - \theta'_A) + j_s^B \eta_B \eta'_B \cos(\theta_B - \theta'_B) + j_a \eta_B \eta'_A \cos(\theta_B - \theta'_A),\tag{5}$$

where $j_0$ represents a uniform inhibitory term; $j_s^A$ and $j_s^B$ measure the amplitude of the symmetric direction-specific connections in map $A$ and map $B$, respectively; and $j_a$ measures the amplitude of asymmetric connections from map $A$ to map $B$. In the Methods, we explain how the dynamics of a network with such recurrent connectivity can reproduce the spatially localized population activity that we observed in the data, and we provide an intuition for the role of the different coupling parameters. A schematic depiction of the network is shown in *Figure 2a*. We parameterized the external input in analogy with the recurrent input:

$$
\begin{aligned}
I^{\text{ext}}(x; t) = \quad & C_0(t) + C_A(t)\eta_A + C_B(t)\eta_B + \\
& \epsilon_A(t)\eta_A \cos(\theta_A - \Phi^{\text{ext}}) + \epsilon_B(t)\eta_B \cos(\theta_B - \Phi^{\text{ext}}),
\end{aligned}\tag{6}
$$

where $C_0$ represents an untuned homogeneous external input, $C_A$ and $C_B$ represent untuned map-specific inputs to maps $A$ and $B$, respectively, $\epsilon_A$ and $\epsilon_B$ represent directionnally tuned inputs to maps $A$ and $B$, respectively, and $\Phi^{\text{ext}}$ is the direction encoded by external inputs. An example of the spatially localized population activity at a given time $t$ visualized in different 2-D subspaces of the 4-D space with coordinates $x = \{\theta_A, \theta_B, \eta_A, \eta_B\}$ is shown in *Figure 2b*, blue and an example of tuning curves from the network's activity is shown in *Figure 2c*.

## Order parameters

The simplicity of the model described by *Equations 1–6* allowed us to derive a low dimensional description of the model dynamics in terms of a few population-level signals denoted as *order parameters*. The order parameters quantify the average population activity $r_0(t)$, and the degree to which the population activity is localized in map A ($r_A(t)$) and B ($r_B(t)$). These parameters are defined by the following equations:

$$
\begin{aligned}
r_0 \quad &= \int dx\rho(x)\, r(\theta_A, \theta_B, \eta_A, \eta_B),\\
r_A \quad &= \int dx\rho(x)\, \eta_A \cos(\theta_A - \psi_A)\, r(\theta_A, \theta_B, \eta_A, \eta_B),\\
r_B \quad &= \int dx\rho(x)\, \eta_B \cos(\theta_B - \psi_B)\, r(\theta_A, \theta_B, \eta_A, \eta_B).
\end{aligned}\tag{7}
$$

Note that $r_A$ and $r_B$ are the first Fourier coefficient of the population rate over the domain $\theta_A$ and $\theta_B$, respectively.

These order parameters can be both computed in the model and from data. Therefore, they can provide us with a simple tool to compare model and data, and to infer model network parameters from data. *Figure 3a* shows the dynamics of the order parameters computed from the data during the delayed reaching task. As expected, we see that during the preparatory period the activity activity is localized in map $A$ ($r_A(t) > 0$) but not in map $B$ ($r_B(t) \sim 0$). Around the time of the go cue, network activity dynamically reorganizes, such as the network becomes now strongly localized in map $B$ ($r_B(t) > 0$), while the degree of modulation in map $A$ slowly decreases towards zero. We can think of map $A$ and map $B$ as two different coordinate systems that the network can use to produce distinct patterns of population activity, that are both localized around the location of the target on the screen, but in different ways. A spatially localized activity profile associated with either $r_A(t) > 0$ or $r_B(t) > 0$ is denoted as a bump. A definition of this term provided in the Methods. At different times, the bump of activity has different shapes – either localised in map $A$, or in map $B$, or in both maps – but its location remains constant. Next, we studied possible mechanisms underlying these observations.

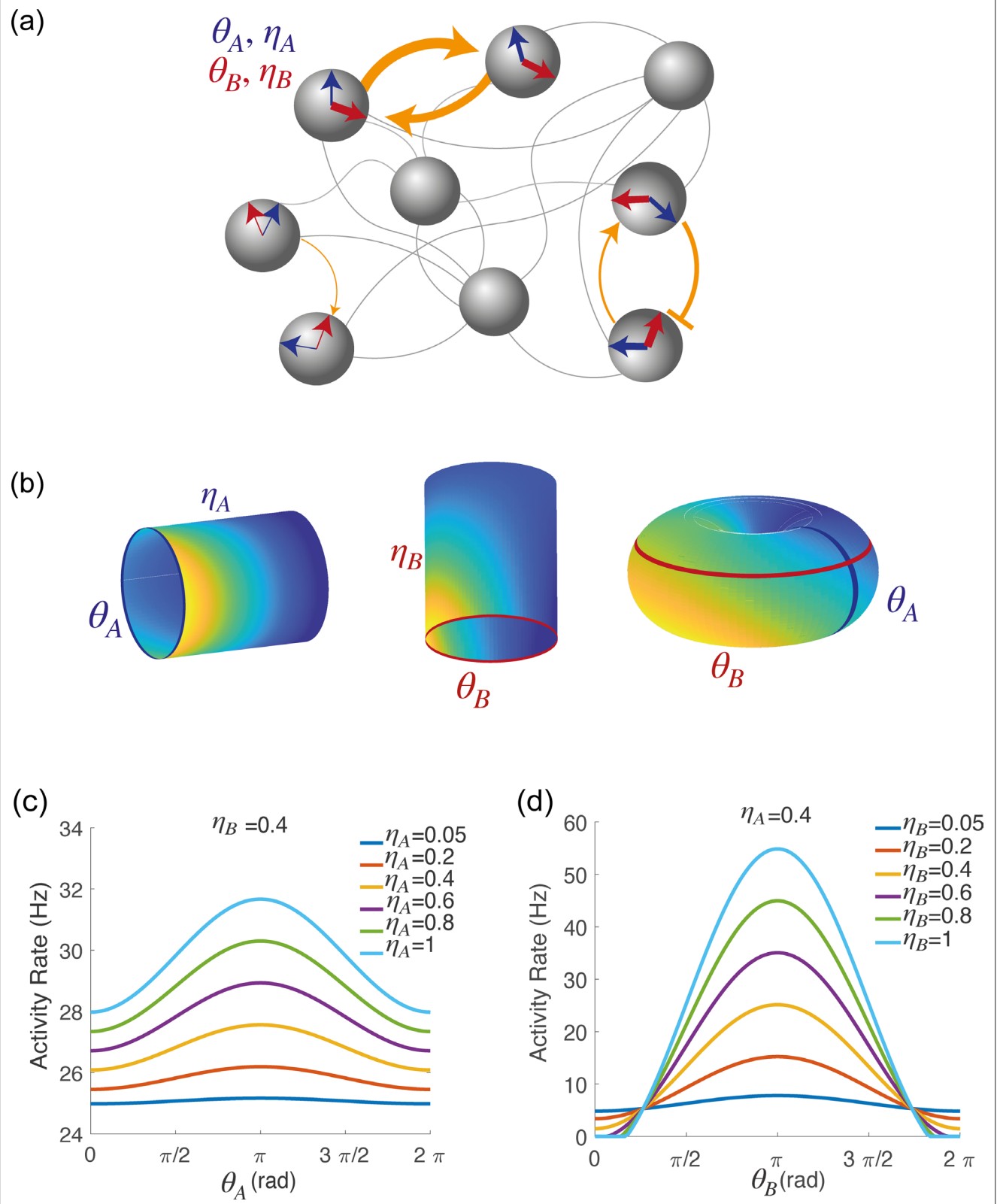

**Figure 2.** Spatially localized activity of the network model. (**a**) Schematic depiction of the network model. Each shaded disk corresponds to one unit in the network. The direction of the blue and red arrows within each disk represents the neurons preferred direction during preparatory ($\theta_A$) and execution ($\theta_B$) epochs, respectively, while the thickness of the arrows represents the degree of participation ($\eta_A, \eta_B$) in these epochs. The orange arrows show the strength of synaptic connectivity (defined by **equation 5**) between units. It contains both a symmetric component, that depends on the distance

*Figure 2 continued on next page*

*Figure 2 continued*

between preferred directions of pre and post-synaptic neurons separately for the two epochs (see strong connections between the two top left neurons), and an asymmetric component that depends on the distance between the preferred preparatory direction of the pre-synaptic neuron and the preferred execution direction of the post-synaptic one (see connections between bottom right neurons). (b) Spatially localized activity of the network at a given time $t$, shown in three different 2-dimensional subspaces of the 4-dimensional selectivity space. Left: activity plotted on map A ($\theta_A, \eta_A$) at fixed $\theta_B, \eta_B$. Center: activity plotted on map B ($\theta_B, \eta_B$) at fixed $\theta_A, \eta_A$. Right: activity plotted as a function of $\theta_A, \theta_B$, at fixed $\eta_A, \eta_B$. (c) Network activity at a given time $t$, plotted as a function of $\theta_A$, at fixed $\theta_B, \eta_B$ and for different values of $\eta_A$. (d) Network activity at a given time $t$, plotted as a function of $\theta_B$, at fixed $\theta_A, \eta_A$ and for different values of $\eta_B$.

## Tuned states in the model: External inputs vs recurrent connectivity

We next investigated the network model to understand the mechanisms underlying the observed dynamics. We first derived equations governing the temporal evolution of the order parameters (see Methods). We then analyzed the stationary solutions of the equations in the absence of tuned inputs ($C_A = C_B = \epsilon_A = \epsilon_B = 0$), in the space of the three parameters defining the couplings strength: $j_0, j_s^A, j_s^B$. We found that, similarly to the ring model (*Ben-Yishai et al., 1995*; *Hansel and Sompolinsky, 1998*), there exists two qualitatively distinct regions in the space of parameters characterizing recurrent connectivity (*Figure 3b*). For weak recurrent connectivity (gray area in *Figure 3b*), the activity in the network is uniform in the absence of tuned inputs. Thus, in this region, tuned network activity must rely on external inputs. For strong recurrent connectivity (white area in *Figure 3b*), network activity is tuned, even in the absence of tuned inputs. In this case, the location of the bump in network activity is determined by initial conditions. These two regimes are separated by a bifurcation line, where the

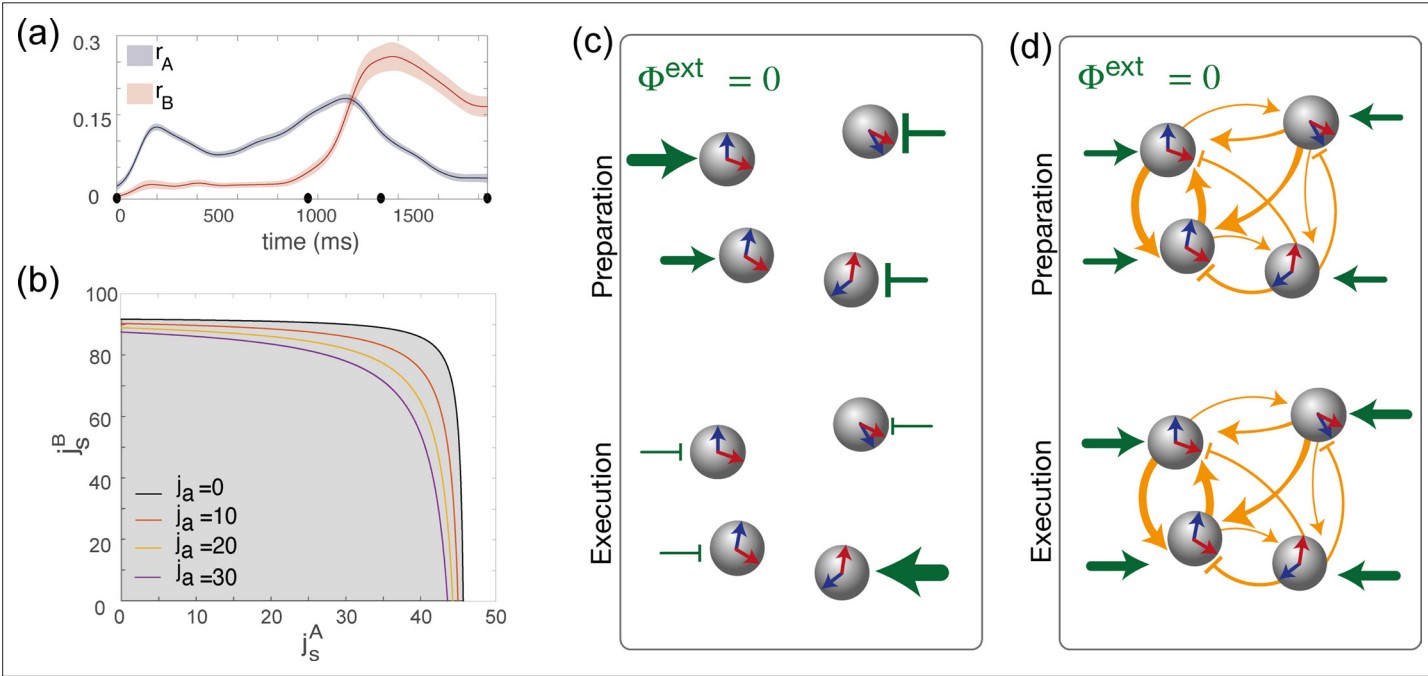

**Figure 3.** Dynamics of the order parameters and phase diagram of the model. (a) Dynamics of the order parameters $r_A$ (degree of spatial modulation of the activity in map A) and $r_B$ (degree of spatial modulation in map B) computed from data (solid line; shaded area: ± SEM across trials). Black dots on the x-axis represent the trial-averaged time of: target onset, go cue, start of movement and end of movement. (b) Phase diagram of the model, shown as a function of the parameters $j_s^A$, $j_s^B$, that describe how strongly maps A and B are embedded in recurrent synaptic connectivity. Different curves correspond to bifurcation lines for different values of the parameter $j_a$, modulating the asymmetric term in synaptic connectivity. The area underneath each line (gray) corresponds to the homogeneous phase; the area beyond each line (white) corresponds to the phase where the network exhibit a localized activity pattern even in absence of tuned external inputs. (c) Weak coupling scenario: Here, the observed dynamics of the order parameters results from external inputs that are tuned to $\theta_A$ during movement preparation and to $\theta_B$ during movement execution. (d) Strong coupling scenario: Here, strong recurrent connections sustain the localized pattern of activity; a change in *untuned* external inputs makes the activity to be localized along map A during preparation, and along map B during preparation.

The online version of this article includes the following figure supplement(s) for figure 3:

**Figure supplement 1.** Results are independent on chosen duration of preparatory and movement-related epochs.

uniform solution becomes unstable due to a Turing instability (**Ben-Yishai et al., 1995**; **Hansel and Sompolinsky, 1998**).

This analysis shows that a network activity profile that is localised in a given map, say map A ($r_A(t) > 0$), can be sustained either thanks to the strong recurrent connectivity (i.e. a strong enough parameter $j_s^A$ in **Equation 5**), or thanks to the external input term proportional to $\epsilon_A(t)$ (**Equation 6**). Hence, our model can potentially generate the observed dynamics of the order parameters (**Figure 3a**) for different choices of recurrent connections and external inputs parameters, ranging in between the following two opposite scenarios:

- Recurrent connections are absent: $j_s^A = j_s^B = j_a = 0$. In this case, the system is simply driven by feedforward inputs. The localized activity is the result of external inputs that selectively excites or inhibits specific neurons during motor preparation and execution, thanks to sufficiently large values of the parameter $r_A(t) > 0$ during preparation and $\epsilon_B(t) > 0$ during execution. This scenario is depicted schematically in **Figure 3c**.
- The network is strongly recurrent, with strongly direction-specific connections, and external inputs are untuned ($\epsilon_A = 0$, $\epsilon_B = 0$). Analytical study of the dynamics (Methods) shows that, in order for such a system to exhibit tuning, the strength of the couplings has to exceed a critical value shown in **Figure 3b**. Moreover, when the synaptic strength exceeds this value, the activity can be localized in map A during preparation and in map B during execution simply as the result of homogeneous external inputs changing their strength, without the need for tuned external inputs to selectively excite/inhibit specific neurons. This scenario is depicted in **Figure 3d**.

We next turn to the question of which of these two scenarios best describes the data.

## Fitting the model to the data

Our next goal is to infer the strength of recurrent connections ($j_0, j_s^A, j_s^B$) and feedforward inputs ($C_0(t), C_A(t), C_B(t), \epsilon_A(t), \epsilon_B(t)$) that best describes the data. To do so, we imposed that our network reproduce the dynamics of the population-level order parameters ($r_0, r_A, r_B$). Specifically, for a given set of recurrent connections and feedforward inputs, we can compute analytically the dynamics of the order parameters (Methods). Based on these calculations, we build an iterative procedure that minimizes the reconstruction error $E_{\text{rec}}$, i.e. the squared difference between the predicted and observed dynamics of the order parameters. The results show that there is a large region of the ($j_0, j_s^A, j_s^B$)-space where the reconstruction error has essentially the same value (**Figure 4—figure supplement 2**). Solutions range from a a network with zero recurrent connections (i.e. a purely feedforward scenario, shown in **Figure 4a-c**) to a network with strong direction-specific connections.

To break the degeneracy of solutions, we added an energetic cost to the reconstruction error. This reflects the idea that a biological network where the computation is only driven by long-range connections from upstream areas is likely to consume more metabolic energy than a network where the computation is processed through shorter-range recurrent connections. Our inference procedure minimizes the cost function:

$$E_{\text{tot}} = \alpha E_{\text{rec}} + E_{\text{ext}},$$

where $E_{\text{ext}}$ is the total magnitude of external inputs (see Methods, **Equation 31**). The hyperparameter $\alpha$ describes the relative strength of these two terms. For very small values of $\alpha$, the algorithm mainly minimizes external inputs, and the resulting reconstruction of the dynamics is poor. For very large values of $\alpha$, multiple solutions are found, that yield similar dynamics (see **Figure 4c** and **Figure 4i**). **Figure 4—figure supplement 1** shows the results we obtained by using intermediate values of $\alpha$, that are small enough to yield a unique solution, but large enough to guarantee a good reconstruction of the dynamics. Interestingly, all solutions cluster in a small region of the parameter space, that is close to the bifurcation surface in such a way that the parameter $j_s^B$ is close to (either larger or smaller) its critical value; $j_s^A$ is much smaller than its critical value, and $j_a$ is zero.

We show the results obtained with two distinct values of $\alpha$ in **Figure 4d** and **Figure 4g**; the corresponding dynamics of the inferred external inputs in **Figure 4e** and **Figure 4f**; and the resulting dynamics of the order parameters from mean field equations in **Figure 4f** and **Figure 4i**. In particular, **Figure 4d** corresponds to a smaller value of $\alpha$, that is we are penalizing more strongly for large external inputs: the resulting coupling parameters are stronger than their critical value. **Figure 4g** corresponds to a larger value of $\alpha$: the strength of the couplings is below the critical line, yielding

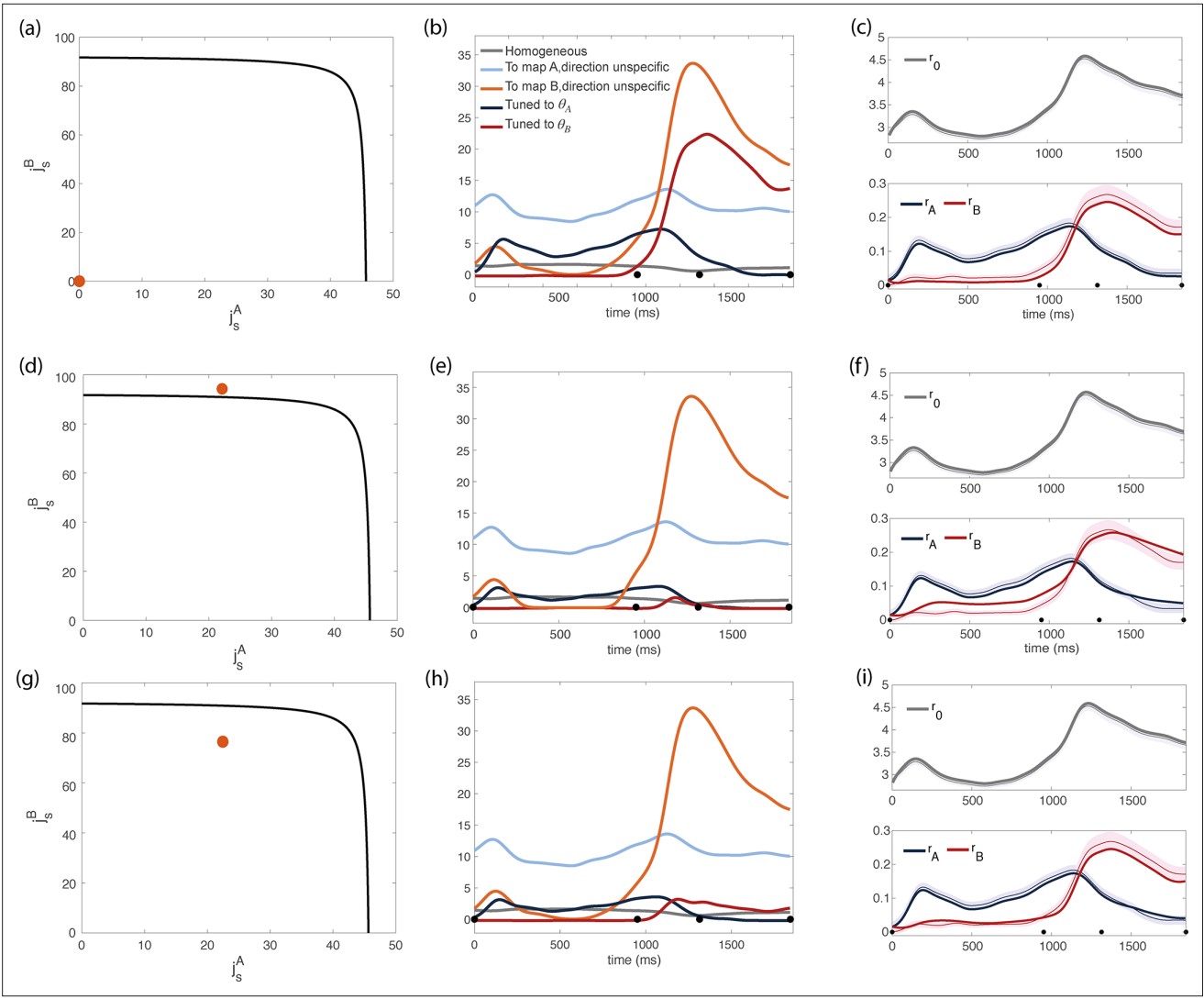

**Figure 4.** Inferred dynamics of the external currents required to sustain the observed dynamics of the order parameters for different sets of couplings parameters $j_s^A, j_s^B, j_a$. (**a–c**) Scenario where neurons are connected by uniform inhibitory connections, in the absence of direction-specific couplings. (**a**) Couplings parameters $j_s^A, j_s^B, j_a$ are set to zero (orange dot on the phase diagram). The black line represents the bifurcation surface in the space $j_s^A, j_s^B$, at $j_a = 0$. (**b**) Dynamics of the external inputs inferred from the data. Gray line: homogeneous (untuned) input; light blue/red line: input that is specific to map A/B, but untuned to direction; input that is directionally tuned, and specific to map A/B. Black dots on the x-axes represent the trial-averaged time of: target onset, go cue, start of movement and end of movement. (**c**) Dynamics of the order parameters $r_0, r_A, r_B$ computed from data (thin line; shaded area: ± SEM across trials) and model (thick line). (**d-i**) Both couplings parameters and external inputs are inferred from data. (**d**) Solution where the coupling parameters are slightly above the bifurcation line ($\alpha = 0.17$). (**d**) Dynamics of the external inputs inferred from the data. Note that tuned inputs are drastically lower than in **b**, especially during movement execution. (**f**) Dynamics of the order parameters $r_0, r_A, r_B$ computed from data (thin line) and model (thick line). (**g**) Solution where the coupling parameters are below but close to the bifurcation line (see *Figure 4—figure supplement 1*, $\alpha = 0.5$). (**h**) Dynamics of the external inputs inferred from the data. (**i**) Dynamics of the order parameters $r_0, r_A, r_B$ computed from data (thin line) and model (thick line).

The online version of this article includes the following figure supplement(s) for figure 4:

**Figure supplement 1.** Value of the couplings parameters $j_s^A, j_s^B$ inferred from data, through minimization of a cost function composed of two terms: $E_{tot} = \alpha E_{rec} + E_{ext}$.

**Figure supplement 2.** Reconstruction error vs cost function.

**Figure supplement 3.** Analysis based on data recorded from monkey Rj.

larger external inputs (*Figure 4h*) and a smaller reconstruction error (*Figure 4i*). The purely feedforward case is added in *Figure 4a-c* for comparison. While a purely feedforward network requires a strong input tuned to map $B$ right before the movement onset (*Figure 4b*), in our solution such input is either absent (*Figure 4e*) or much weaker than the corresponding untuned input (*Figure 4h*). Our analysis therefore suggests that recurrent connectivity plays a major role in maintaining the degree of spatial modulation observed in the data.

Importantly, the same analysis applied to a second dataset recorded from a different macaque monkey performing the same task yielded qualitatively similar results (see *Figure 4—figure supplement 3*).

## The model generates realistic neural and muscle activity

We have shown that the model is able to reproduce the dynamics of the order parameters computed from data. Here, we ask how the activity of single neurons in our model compares to data, and whether a readout of neural activity can generate realistic patterns of muscle activity. We simulated the dynamics of a network of 16,000 neurons. To each neuron $i$, we assigned the coordinates $\theta_A^{(i)}, \theta_B^{(i)}, \eta_A^{(i)}, \eta_B^{(i)}$ so as to match the empirical distribution of coordinates (*Equation 3* and *Figure 5—figure supplement 3a*). We added a noise term to the total current that each neuron is subject to, modelled as an Ornstein-Uhlenbeck process (see Methods, *Equation 33*).

First, we checked that the dynamics of the order parameters – previously computed by numerically integrating the analytical equations – is also correctly reconstructed by simulations (*Figure 5—figure supplement 1b*). *Figure 5—figure supplement 1* shows the case of a network with couplings parameters stronger than their critical value, where the dynamics is dominated by recurrent currents and is very sensitive to noise The location of the bump undergoes a small diffusion around the value predicted by the mean field analysis (*Figure 5—figure supplement 1c* and *Figure 5—figure supplement 1d*).

Then, we computed tuning curves during the preparatory and execution epochs from simulations, and estimated the values $\theta_A^{(i)}, \theta_B^{(i)}, \eta_A^{(i)}, \eta_B^{(i)}$ from the location and the amplitude of the tuning functions. The reconstructed values of neurons tuning parameters are consistent with the values initially assigned to them when we built the network (*Figure 5—figure supplement 2b*). We noticed (*Figure 5—figure supplement 2c*) that the quality of the cosine fit strongly correlates with the magnitude of $\eta_A, \eta_B$, as we see in the data (*Figure 1—figure supplement 2e-f*): for neurons with a low value of $\eta_A, \eta_B$ the tuning functions poorly resemble a cosine. Overall, tuning curves from simulations strongly resemble the ones from data (see *Figure 5—figure supplement 3*, and Discussion).

Next, we showed that the network activity closely resembles the one from recordings. At the level of the population, we used canonical correlation analysis to identify common patterns of activity between simulations and recordings, and to show that they strongly correlate (see *Figure 5a* and Methods). At the level of single units, for each neuron in the data we selected a neuron in the model network with the closest value of $\theta_A, \theta_B, \eta_A, \eta_B$ variables. A side-by-side comparison of the time course of responses shows a good qualitative agreement (*Figure 5c*). We also noted that both the activity from recordings and from simulations present sequential activity and rotational dynamics (*Figure 5—figure supplement 4*). Finally, a linear readout of the activity from simulations can generate realistic patterns of muscle activity, which closely match electromyographic (EMG) signals recorded from Monkey Bx during a center-out reaching movement (*Figure 5b*).

So far in this section, we discussed results from a network with strong couplings; *Figure 5—figure supplement 5* shows that the dynamics of a purely feedforward network is almost indistinguishable from from the strongly recurrent case - although canonical correlations between the activity of a purely feedforward network and the data are slightly lower than the ones for networks with strong recurrent couplings, during movement execution (average canonical correlation of 0.74 for the purely feedforward network, and 0.82 for the strongly recurrent one). We also noticed that networks with coupling parameters below the bifurcation surface are robust to noise in the $\theta_A^{(i)}, \theta_B^{(i)}$ variables. Conversely, the solutions above the bifurcation surface that do not require tuned external inputs are very sensitive to such noise, and the location of the bump of the activity drifts towards a few discrete attractors. When the noise is weak, the drift happens on a time scale that is much larger that the time of movement execution; the larger the level of the noise, the faster the drift.

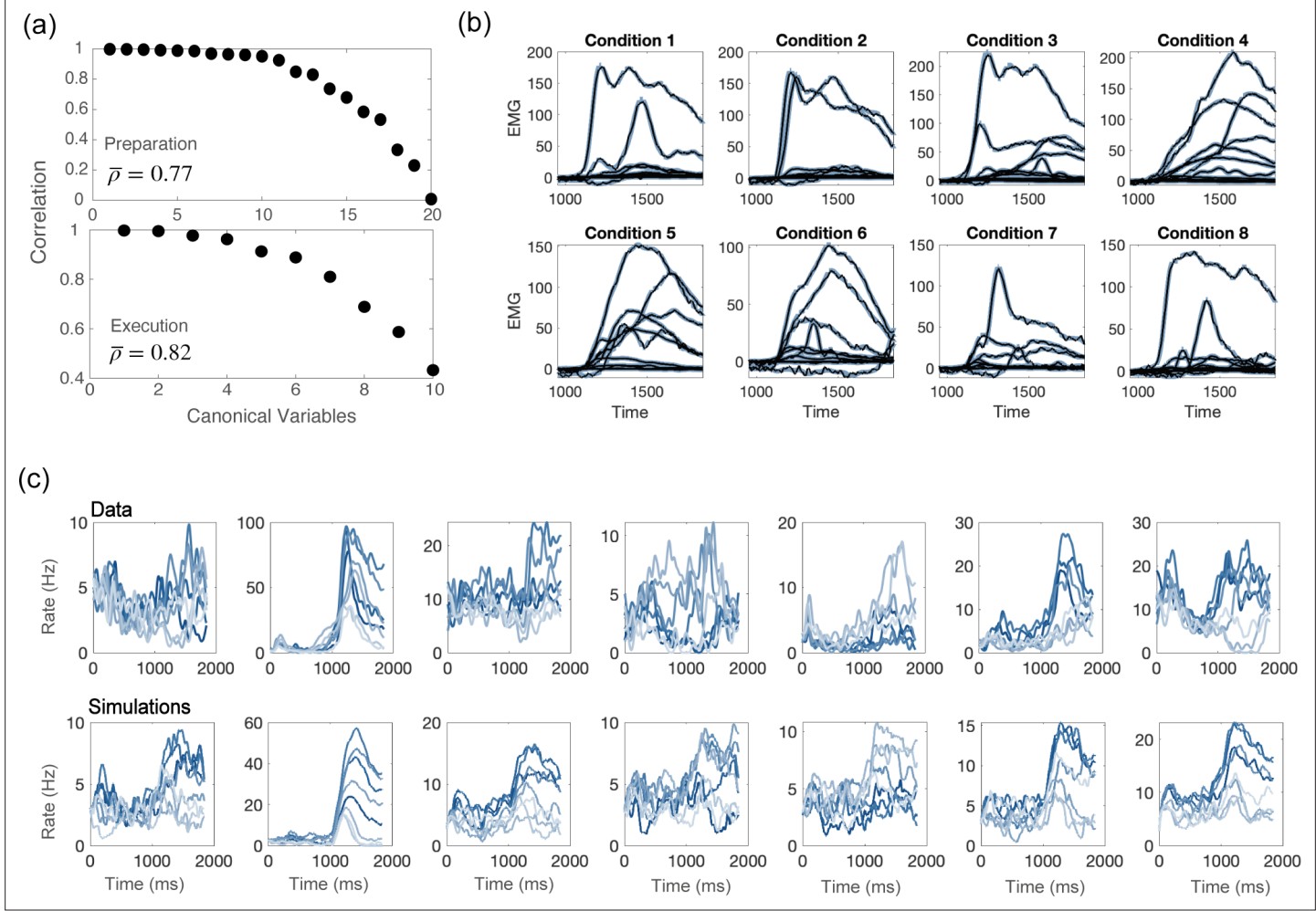

**Figure 5.** Comparisons between model and data, for parameters of *Figure 4d*. (**a**) CCA analysis to compare the population activity from simulations and from recordings. First, we projected the activity onto the PCA dimensions that captured 90% of the activity variance, for both the data and the simulations; then, we applied CCA to look for common patterns in the activity matrices from data and simulations. Top: canonical correlations, activity during preparation. Bottom: canonical correlations, activity during execution. (**b**) Blue lines: electromyographic (EMG) signals from 13 muscles, recorded during a center-out reaching movement. Each panel corresponds to a different condition (location of the target on the screen). Black lines: patterns of muscle activity predicted by a linear readout of the activity of 1000 units drawn at random from the 16000 units in the network model (cross- validated NMSE = 0.0066). (**c**) For each neuron in the data, we chose the corresponding one from simulations with the closest value of parameters $\theta_A, \theta_B, \eta_A, \eta_B$. Top row: examples of trial averaged activity from data; bottom row: corresponding neurons from simulations. Different shades of blue correspond to the 8 different conditions.

The online version of this article includes the following figure supplement(s) for figure 5:

**Figure supplement 1.** Dynamics of the order parameters from simulations, for the coupling parameters indicated in red in panel (**a**), above but close to the bifurcation line.

**Figure supplement 2.** Network architecture for simulations.

**Figure supplement 3.** Examples of tuning curves during the preparatory (blue) and movement-related (red) epochs computed from simulations (**a**) and from data (**b**).

**Figure supplement 4.** Rotational dynamics and sequential activity.

**Figure supplement 5.** Simulations of a network with no recurrent connections.

## PCA subspaces dedicated to movement preparation and execution

In the previous sections, we considered the low-dimensional description of the dynamics given by the order parameters. Here, we use the commonly used principal component analysis (PCA) to compare the dimensionality of data and network model. In particular, we ask whether our formalism can explain the orthogonality of the preparatory and movement-related linear subspaces. As previously reported

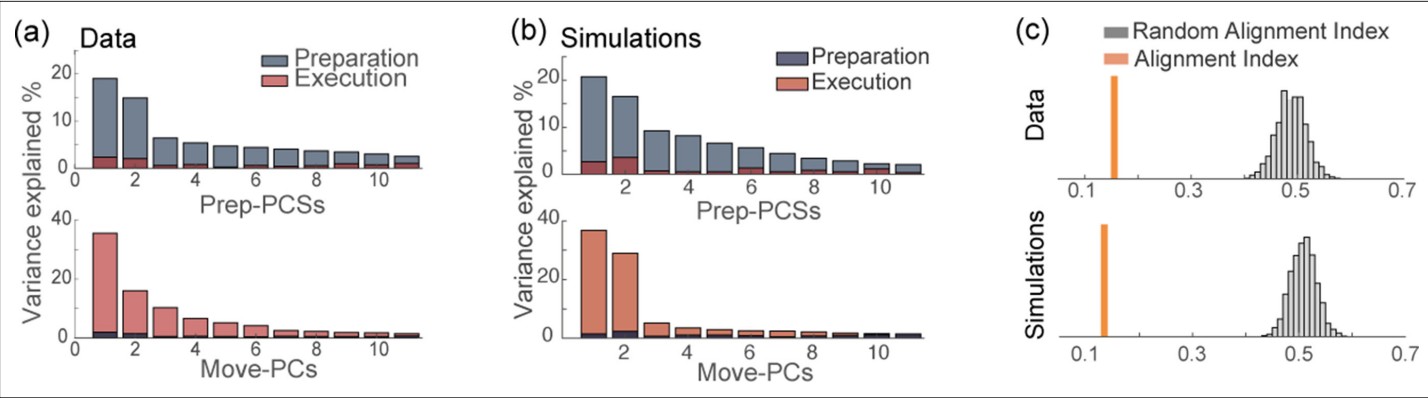

**Figure 6.** Orthogonality of the preparatory and movement-related subspaces. (**a**) Percentage of variance of the preparatory (blue) and movement-related (red) activity from data explained by the first 11 principal components calculated from preparatory (top) and movement-related (bottom) trial-averaged activity. (**b**) As in **a**, activity from simulations. (**c**) The alignment index quantifies the degree of orthogonality between two subspaces. Top: alignment index between the preparatory and movement-related activities computed from data, compared to the randomized test (random alignment index, distribution in light gray and average in dark grey). Bottom: alignment index computed from simulations.

The online version of this article includes the following figure supplement(s) for figure 6:

**Figure supplement 1.** PCA analysis of network dynamics in various conditions.

**Figure supplement 2.** Network with uniform degrees of tuning, $\eta_A = \eta_B = 1$ for all neurons.

in the literature *Kaufman et al., 2014*; *Elsayed et al., 2016*, the top panel in *Figure 6a* shows that the top principal components of the preparatory activity explain most of the variance of the preparatory activity, but very little variance of the movement-related activity; vice versa, the top movement-related principal components explain very little variance of the preparatory activity (*Figure 6a*, bottom panel). The activity of our network model also displays this property (*Figure 6b*). We also quantified the level of orthogonality of the two subspaces using the alignment index as defined in *Elsayed et al., 2016*, that measures the percentage of variance of each epoch's activity explained by both sets of PCs. *Figure 6c* shows that the alignment index is much smaller than the one computed between two subspaces drawn at random (random alignment index, explained in the Methods), in both data and network simulations. Note that model with uniform degrees of participation ($\eta_A = \eta_B = 1$) displays a higher overlap between the preparatory and movement-related subspaces, as shown in *Figure 6—figure supplement 2c*. In the Methods section, we explain how the dimensionality of the activity depends on the connectivity of the network.

## Discussion
### Orthogonal spaces dedicated to movement preparation and execution
Studies on the dynamics of motor cortical activity during delayed reaching tasks have shown that the primary motor cortex employs an 'orthogonal but linked strategy' (*Kaufman et al., 2014*; *Elsayed et al., 2016*) to coordinate planning and execution of movements.

In this work, we explored the hypothesis that this strategy emerges as result of a specific recurrent functional architecture, in which synaptic connections store information about two distinct patterns of activity that underlie movement preparation and movement execution. We built a network model in which neurons are characterized by their selectivity properties in both preparatory and movement execution epochs (preferred direction and degree of participation), and in which synaptic connectivity is shaped by these selectivity properties in a Hebbian-like fashion.

A strong correlation between the selectivity properties of the preparatory and movement-related epochs will produce strongly correlated patterns of activity in these two intervals and a strong overlap between the respective PCA subspaces. We inferred the distribution of these tuning features from data and showed that the correlation between the preparatory and movement-related patterns of activity is small enough to allow for almost orthogonal subspaces, which is thought to be important for the preparatory activity not to cause premature movement. At the same time, the correlation is

non-zero, and that allows the activity to flow from the preparatory to the movement-related subspaces with minimal external inputs, as summarized in the next section.

## Interplay between external and recurrent currents

We analytically described the temporal evolution of the population activity in the low-dimensional space defined by maps A and B in terms of a few order parameters, which can be easily computed from data. Different combinations of the strength of direction-specific recurrent connections and of tuned external inputs allow the model to accurately reproduce the dynamics of the order parameters from data. We argue that solutions that require less inputs from areas upstream of the motor cortex are favorable in terms of metabolic energy consumption. With the addition of a cost proportional to the magnitude of external inputs, we find solutions where recurrent connections are strong and direction specific. In the resulting scenario, during movement preparation, an external input tuned to map A sustains a population-level activity localized in map A, and pins the location of the peak of activity. During movement execution, the localized activity is sustained mostly by recurrent connectivity; the correlation between preferred directions $\theta_A$ and $\theta_B$ allows the activity in map B during movement execution to be localized around the same location as it was in map A during movement preparation. The inferred strength of recurrent connectivity is close to the critical value above which the recurrent network can generate localized patterns of activity in the absence of tuned inputs. Solutions well beyond the bifurcation line require an implausible fine tuning of the recurrent connections, as heterogeneity in the connectivity causes a systematic drift of the encoded direction of motion on a typical time scales of seconds – the larger the structural noise in the couplings, the faster the drift, as has been extensively studied in continuous attractor models (*Tsodyks and Sejnowski, 1995*; *Zhang, 1996*; *Renart et al., 2003*; *Itskov et al., 2011*; *Seeholzer et al., 2019*; *Compte et al., 2000*). It has been shown that homeostatic mechanisms could compensate for the heterogeneity in cellular excitability and synaptic inputs to reduce systematic drifts of the activity (*Renart et al., 2003*) and that short-term synaptic facilitation in recurrent connections could also significantly improve the robustness of the model (*Itskov et al., 2011*) – even when combined with short-term depression (*Seeholzer et al., 2019*). While a full characterization of our model in the presence of structural heterogeneity is beyond the scope of this work, we note that solutions close but below the bifurcation line are stable with respect to perturbations in the couplings. In this case, tuned inputs are present also during movement execution, but their magnitude is much weaker than the untuned ones: direction-specific couplings are strong and amplify the weak external inputs tuned to map B, therefore playing a major role into shaping the observed dynamics during movement execution.

Our prediction that external inputs are direction-specific during movement preparation but mostly non-specific during movement execution needs to be tested experimentally. Interestingly, it agrees with several previous studies on the activity of the primary motor cortex during limb movement. In particular, *Kaufman et al., 2016* showed that changes in neural activity that characterize the transition from movement preparation to execution reflect when movement is made but are invariant to movement direction and type, in monkeys performing a reaching task; *Inagaki et al., 2022* detected a large, movement non-specific thalamic input to the cortex just before movement onset, in mice performing a licking task. The authors of *Nashef et al., 2019* used high-frequency stimulation to interfere with the normal flow of information through the cerebellar-thalamo-cortical (CTC) pathway in monkeys performing a center-out reaching task. This perturbation produced reversible motor deficits, preceded by substantial changes in the activity of motor cortical neurons around movement execution. Interestingly, the spatial tuning of motor cortical cells was unaffected, and their early preparatory activity was mostly intact. These results are in line with our prediction, if we interpret the condition-invariant inputs that we inferred during movement execution as thalamic inputs that are part of the CTC pathway. We speculate that the direction-specific inputs that we inferred during movement preparation have a different origin. Further simultaneous recordings of M1 and upstream regions, as well as measures of synaptic strength between motor cortical neurons, will be necessary to test our predictions.

## Cosine tuning

While cosine tuning functions have been extensively used to describe the firing properties of motor cortical neurons (e.g. *Georgopoulos et al., 1982*), and they were also hypothesized to be the optimal

tuning profile to minimize the expected errors in force production (*Todorov, 2002*), more recent work has emphasized that tuning functions in the motor cortex present heterogeneous shapes. Specifically, the authors of *Lalazar et al., 2016* showed that tuning functions are well fitted by a sum of a cosine-modulated component, and an unstructured component, often including terms with higher spatial frequency. They also argued that the unstructured component is key for a readout of motor cortical activity to reproduce EMG activity. In this work, we showed that a noisy recurrent network model that is based on cosine tuning reproduces well the $R^2$ coefficient of the cosine fit of tuning curves from data (see a comparison between *Figure 1—figure supplement 2e-f* and *Figure 5—figure supplement 2c*). Moreover, tuning curves from simulations present heterogeneous shapes (*Figure 5—figure supplement 3*), including bimodal profiles, especially for neurons with low values of $\eta_A, \eta_B$. Indeed, we showed that the $R^2$ of the cosine fit strongly correlates with the variables $\eta$, both in the data and in the model. Neurons with a low degree of participation have low $R^2$ coefficient of the cosine fit, and present heterogeneous tuning curves. We also showed that a linear readout of the activity of our model can very accurately reconstruct EMG activity. Finally, our model could easily be extended to incorporate other tuning functions. While we leave the analysis of a model with an arbitrary shape of the tuning function to future work, we don't expect our results to strongly depend on the specific shape of the tuning function.

## Comparison with the ring model

The idea that the tuning properties of motor cortical neurons could emerge from direction-specific synaptic connections goes back to the work of *Lukashin and Georgopoulos, 1993*. However, it was with the theoretical analysis of the so called ring model (*Amari, 1977*; *Ben-Yishai et al., 1995*; *Somers et al., 1995*; *Hansel and Sompolinsky, 1998*) that localized patterns of activity were formalized as attractor states of the dynamics in networks with strongly specific recurrent connections. Related models were later used to describe maintenance of internal representations of continuous variables in various brain regions (*Zhang, 1996*; *Redish et al., 1996*; *McNaughton et al., 1996*; *Seung et al., 2000*; *Tsodyks, 1999*; *Camperi and Wang, 1998*; *Compte et al., 2000*; *Samsonovich and McNaughton, 1997*; *Stringer et al., 2002*; *Burak and Fiete, 2009*) and were extended to allow for storage of multiple continuous manifolds (*Battaglia and Treves, 1998*; *Romani and Tsodyks, 2010*; *Monasson and Rosay, 2015*) to model the firing patterns of place cells the hippocampus of rodents exploring multiple environments. While our formalism builds on the same theoretical framework of these previous works, we would like to stress two main differences between our model and the ones previously considered in the literature. First, we studied the dynamic interplay between fluctuating external inputs and recurrent currents, that causes the activity to flow from the preparatory map to the movement-related one and, consequently, neurons tuning curves and order parameters to change over time, while at the population level the pattern of activity remains localized around the same location. Then, we introduced an extra dimension representing the degree of participation of single neurons to the population pattern of activity; this is an effective way to introduce neuron-to-neuron variability in the responses, which decreases the level of orthogonality between the preparatory and movement-related subspaces, and yields tuning curves whose shape resembles the one computed from data – in contrast with the classic ring model, where all tuning curves have the same shape.

## Representational vs dynamical system approaches

There has been a debate whether neuronal activity in motor cortex is better described by so-called 'representational models' or dynamical systems models (*Churchland and Shenoy, 2007*; *Michaels et al., 2016*). Representational models (*Michaels et al., 2016*; *Inoue et al., 2018*) are models in which neuronal firing rates are related to movement parameters (*Evarts, 1968*; *Georgopoulos et al., 1982*; *Georgopoulos et al., 1984*; *Paninski et al., 2004*; *Moran and Schwartz, 1999*; *Smith et al., 1975*; *Hepp-Reymond et al., 1978*; *Cheney and Fetz, 1980*; *Kalaska et al., 1989*; *Taira et al., 1996*; *Cabel et al., 2001*). Dynamical systems models (*Sussillo et al., 2015*) are instead recurrent neural networks whose synaptic connectivity is trained in order to produce a given pattern of muscle activity. Such models have been argued to reproduce better the dynamical patterns of population activity in motor cortex (*Michaels et al., 2016*).

In our model, firing rates are described by a system of coupled ODEs, and the synaptic connectivity is built from the neuronal tuning properties, in the spirit of the classical ring model discussed above,

and of the model introduced by *Lukashin and Georgopoulos, 1993*. Our model is thus a dynamical system where kinematic parameters can be decoded from the population activity.

Importantly, our model is constrained to reproduce the dynamics of a few order parameters that are a low-dimensional representation of the activity of recorded neurons. In contrast to kinematic-encoding models, our model can recapitulate the heterogeneity of single-unit responses. Moreover, as in trained recurrent network models, a linear readout of the network activity can reproduce realistic muscle signals.

The advantage of our model with respect to a trained RNN is that it yields a low-rank connectivity matrix which is simple enough to allow for analytical tractability of the dynamics. The model can be used to test specific hypotheses on the relationship between network connectivity, external inputs and neural dynamics, and on the learning mechanisms that may lead to the emergence of a given connectivity structure. The model is also helpful to illustrate the problem of degeneracy of network models. An interesting future direction would be to compare the connectivity matrices of trained RNNs and of our model.

### Extension to modeling activity underlying more complex tasks

Neurons directional tuning properties have been shown to be influenced by many contextual factors that we neglected in our analysis (*Hepp-Reymond et al., 1999*; *Muir and Lemon, 1983*), to depend on the acquisition of new motor skills (*Paz et al., 2003*; *Li et al., 2001*; *Wise et al., 1998*) and other features of movement such as the shoulder abduction/adduction angle even for similar hand kinematic profiles (*Scott and Kalaska, 1997*; *Scott et al., 2001*), or the speed of movement for similar trajectories (*Churchland and Shenoy, 2007*). A large body of work (*Scott et al., 2001*; *Ajemian et al., 2000*; *Gribble and Scott, 2002*; *Hepp-Reymond et al., 1999*; *Todorov, 2000*; *Holdefer and Miller, 2002*; *Sergio et al., 2005*) has shown that the activity in the primary motor cortex covaries with many parameters of movement other than the hand kinematics – for a review, see *Scott, 2003*. More recent studies *Michaels et al., 2016*; *Russo et al., 2018*; *Sergio et al., 2005*; *Schroeder et al., 2021* have also suggested that the largest signals in motor cortex may not correlate with task-relevant variables at all.

Our model can be extended to more realistic scenarios in several ways. A simplifying assumption we made is that the task can be clearly separated into a preparatory phase and one movement-related phase. A possible extension is one where the motor action is composed of a sequence of epochs, corresponding to a sequence of maps in our model. It will be interesting to study the role of asymmetric connections for storing a sequence of maps. Such a network model could be used to study the storage of motor motifs in the motor cortex (*Logiaco et al., 2021*); external inputs could then combine these building blocks to compose complex actions. Moreover, incorporating variability in the degree of symmetry of the connections could allow to model features that are not currently included, such as the speed of movement.

In summary, we proposed a simple model that can explain recordings during a reaching task. It provides a scaffold upon which more sophisticated models could be built, to explain neural activity underlying more complex tasks.

## Methods

### Animals

*Figures 1–6* are based on electrophysiological recordings from the primary motor cortex of Macaque monkey Rk (141 units, 391 trials), while in *Figure 4—figure supplement 3* we analyzed recordings from Macaque monkey Rj (57 units, 191 trials). For details on the electrophysiology and on the multi-electrode array implants, see *Russo et al., 2018*.

### Circular statistics

To define the statistical measures of angular variables (*Fisher, 1995*; *Berens, 2009*), let us consider a set of angles $\{\theta_i\}_i$, and visualize them as vectors on the plane:

$$\Theta_i = \begin{bmatrix} \cos\theta_i \\ \sin\theta_i \end{bmatrix}. \tag{8}$$

The mean vector is then defined as

$$\overline{\Theta} = \frac{1}{N}\sum_i \Theta_i \tag{9}$$

from which we can get the mean angular direction $\overline{\theta}$ using the four quadrant inverse tangent function. The length of the mean resultant vector,

$$R = \|\overline{\Theta}\|, \tag{10}$$

is a measure of how concentrated the data sample is around the mean direction: the closer $R$ is to 1, the more concentrated the data is. We quantify the spread in a data set through the circular variance given by

$$S = 1 - R, \tag{11}$$

which ranges from 0 to 1. To compute the correlation between two sets of angular variables, $\{\theta_i\}_i$ and $\{\phi\}_i$, we used the correlation coefficient (*Jammalamadaka and SenGupta, 2001*) defined as:

$$\rho = \frac{\sum_i \sin(\theta_i - \overline{\theta})\sin(\phi_i - \overline{\phi})}{\sqrt{\sum_i \sin^2(\theta_i - \overline{\theta})\sin^2(\phi_i - \overline{\phi})}}, \tag{12}$$

where $\overline{\theta}$ and $\overline{\phi}$ are the mean angular directions of the two sets.

## Preparatory and movement-related epochs

In our analysis, we characterized neurons' tuning properties during movement preparation and execution. The choice of considering only two epochs is an approximation based on the observation that neurons preferred directions are more conserved within the delay period alone and within movement execution alone than across periods (*Figure 1—figure supplement 1*). In *Figure 1—figure supplement 1*, we considered three temporal epochs: the delay period (blue lines), movement execution (red) and the whole duration of the task (grey). We binned each epoch into $N_W$ time bins of length 160 ms: the preparatory and execution epochs consist of $N_W = 3$ bins each, while the whole task consists of $N_W = 7$ bins. For each neuron, we first computed its preferred direction in each time bin, by fitting the trial-averaged and time-averaged firing rate with a cosine function. Then, for each epoch, and for each neuron, we computed the circular variance of preferred directions across the $N_W$ bins. The cumulative distribution of circular variances is shown in *Figure 1—figure supplement 1a*: the variability of preferred direction within the delay epochs and within movement execution is non-zero, although their distribution is skewed towards zero.

For each epoch, we computed the median variability in preferred direction and used a bootstrap analysis to check its statistical significance. Let us consider the matrix of *single trial* activity of a given neuron, for a specific condition, and during a given epoch; the size of the matrix is $N_W \times n$, where $n$ is the number of trials per condition (in the data, $n$ ranges between 41 and 47). We repeat the same procedure for all of the 8 conditions, and z-scored the 8 activity matrices across conditions (this is done because the average activity changes across time bins, and in the following we will shuffle time bins). To check if tuning curves are exactly conserved across time-bins, we generated $B = 1000$ bootstrap samples of the activity matrix for a given condition, where each entry is chosen at random (with repetitions) between the possible $N_W \times n$ entries of the original matrix. For each bootstrap sample matrix, we computed the variance of preferred direction as we did for the original matrix. This is repeated for all neurons, all epochs, all conditions. The cumulative distribution of all variances of preferred direction is shown in *Figure 1—figure supplement 1b*. Then, for each bootstrap sample, we computed the corresponding median variance of preferred direction; the histogram is shown in *Figure 1—figure supplement 1c*. From *Figure 1—figure supplement 1c*, it is obvious that the median variance of preferred directions from the data is significantly larger than the one from the bootstrap samples. This suggests that neurons do change their preferred direction within epochs, even though major changes happen between epochs.

## Analysis of the model

We studied the rate model with couplings defined by (5) with two complementary approaches. First, an analytic approximation based on mean-field arguments and valid in the limit of large network size allowed us to derive a low-dimensional description of the network dynamics in terms of a few latent variables, and to fit the model parameters to the data. Next, we checked that simulations of the dynamics of a network of $N = 10^4$ neurons reproduce the results that we derived in the limit $N \to \infty$.

The mean-field equations are derived following the methods introduced in *Ben-Yishai et al., 1995*; *Romani and Tsodyks, 2010*. In the limit where the number of neurons is large, the average activity of a neuron with coordinates $\theta_A, \theta_B, \eta_A, \eta_B$ is described by *Equations 1; 2*, where we have defined $x = \{\theta_A, \theta_B, \eta_A, \eta_B\}$. In order to derive a lower dimensional description of the dynamics, we rewrite (1) in terms of the average activity rate (14), and of the second Fourier components of the activity rate modulated by $\eta_{A/B}$:

$$
\begin{aligned}
Z_A &\equiv \int dx\, \rho(x)\, \eta_A\, r(\theta_A, \theta_B, \eta_A, \eta_B)\, e^{i\theta_A} \equiv r_A e^{i\psi_A} \\
Z_B &\equiv \int dx\, \rho(x)\, \eta_B\, r(\theta_A, \theta_B, \eta_A, \eta_B)\, e^{i\theta_B} \equiv r_B e^{i\psi_B}.
\end{aligned}
\tag{13}
$$

The phase $\psi_{A(/B)}$ is defined so that the parameter $r_{A(/B)}$ is a real nonnegative number, yielding *Equation 14* for $r_{A(/B)}$, and the following equation for the phase $\psi_{A(/B)}$, representing the position of the peak of the activity profile in map $A(/B)$:

$$
\begin{aligned}
0 &= \int dx\, \rho(x)\, \eta_A\, \sin(\theta_A - \psi_A)\, r(\theta_A, \theta_B, \eta_A, \eta_B) \\
0 &= \int dx\, \rho(x)\, \eta_B\, \sin(\theta_B - \psi_B)\, r(\theta_A, \theta_B, \eta_A, \eta_B).
\end{aligned}
\tag{14}
$$

From (1), we see that the order parameters evolve in time according to the following set of equations,

$$
\begin{aligned}
\tau \frac{d}{dt} r_0(t) &= -r_0(t) + \int dx\, \rho(x)\, [I^{\text{tot}}(\theta_A, \theta_B, \eta_A, \eta_B; t)]_+ \\
\tau \frac{d}{dt} r_A(t) &= -r_A(t) + \int dx\, \rho(x)\, \eta_A\, \cos(\theta_A - \psi_A(t))\, [I^{\text{tot}}(\theta_A, \theta_B, \eta_A, \eta_B; t)]_+ \\
\tau \frac{d}{dt} r_B(t) &= -r_B(t) + \int dx\, \rho(x)\, \eta_B\, \cos(\theta_B - \psi_B(t))\, [I^{\text{tot}}(\theta_A, \theta_B, \eta_A, \eta_B; t)]_+ \\
\tau r_A(t) \frac{d}{dt} \psi_A(t) &= \int dx\, \rho(x)\, \eta_A\, \sin(\theta_A - \psi_A(t))\, [I^{\text{tot}}(\theta_A, \theta_B, \eta_A, \eta_B; t)]_+ \\
\tau r_B(t) \frac{d}{dt} \psi_B(t) &= \int dx\, \rho(x)\, \eta_B\, \sin(\theta_B - \psi_B(t))\, [I^{\text{tot}}(\theta_A, \theta_B, \eta_A, \eta_B; t)]_+,
\end{aligned}
\tag{15}
$$

where the total input (2) is rewritten as:

$$
\begin{aligned}
I^{\text{tot}}(x; t) = \ & I^{\text{ext}}(\theta_A, \theta_B, \eta_A, \eta_B; t) + j_0 r_0(t) + \\
& \sum_{\nu = A, B} j_s^\nu \eta_\nu \cos\left(\theta_\nu - \psi_\nu(t)\right) r_\nu + j_a \eta_B \cos\left(\theta_B - \psi_A(t)\right) r_A(t).
\end{aligned}
\tag{16}
$$

## Stationary states for homogeneous external inputs

We first characterize the model by studying the fixed-points of the network dynamics when subject to a constant and homogeneous external input, and focusing on the scenario where the joint distribution (3) of the $\theta_A, \theta_B$ is of the form

$$
\rho_d(\theta_A, \theta_B) = \frac{1}{4\pi^2} \left[1 + x \cos(\theta_A - \theta_B)\right],
\tag{17}
$$

which fits the empirical distribution of the data well for $x = 2/3$ (*Figure 1—figure supplement 2*). For now, we leave the distributions $\rho_{pA}(\eta_A)$ and $\rho_{pB}(\eta_B)$ unspecified. We first consider the case where the external input to the network is a constant that is independent of $\theta_A, \theta_B$:

$$
I^{\text{ext}}(x) = C_0 + \eta_A C_A + \eta_B C_B.
\tag{18}
$$

The stationary solutions of (1,16) are of the form:

$$
r(x) = \left[I^{\text{tot}}\right]_+ = \left[I_0 + I_A \cos(\theta_A - \psi_A) + I_B \cos(\theta_B - \psi_B)\right]_+,
\tag{19}
$$

where we have defined

$$
\begin{aligned}
I_0 &= C_0 + \eta_A C_A + \eta_B C_B + j_0 r_0 \\
I_A &= j_s^A \eta_A r_A \\
I_B &= j_s^B \eta_B r_B + j_a \eta_B r_A.
\end{aligned}
\tag{20}
$$

Here, $\{r_0, r_A, r_B\}$ are solutions of the system (15) with the left hand side set to zero. The second term on the r.h.s in the last equation of (20) is obtained from (16) by observing that in the stationary state $\psi_A = \psi_B$ if either $j_a > 0$ or if $\theta_A$ and $\theta_B$ are correlated (as we are assuming here). As in **Ben-Yishai et al., 1995**; **Hansel and Sompolinsky, 1998**, we can distinguish broad from narrow activity profiles. The term *broad activity profile* refers to the scenario where the activity of all the neurons is above threshold, the dynamics is linear and the stationary state reduces to:

$$
r(x) = I_0 + I_A \cos(\theta_A - \psi_A) + I_B \cos(\theta_B - \psi_B).
\tag{21}
$$

By inserting the above equation in (14), we find that the only solution is homogeneous over the maps $\theta_A$ and $\theta_B$:

$$
\begin{aligned}
r_0 &= \frac{C_0 + \langle \eta_A \rangle C_A + \langle \eta_B \rangle C_B}{1 - j_0}, \\
r_A &= 0, \\
r_B &= 0, \\
r(\eta_A, \eta_B) &= C_0 + \eta_A C_A + \eta_B C_B + j_0 r_0,
\end{aligned}
\tag{22}
$$

where the notation $\langle . \rangle$ represents an average over the measure $d\mu$, i.e.

$$
\langle \eta_A \rangle = \int_0^1 d\eta \, \rho_{pA}(\eta) \, \eta, \quad \langle \eta_B \rangle = \int_0^1 d\eta \, \rho_{pB}(\eta) \, \eta.
$$

First, we notice that a nonzero homogeneous state is present if

$$
\frac{C_0 + \langle \eta_A \rangle C_A + \langle \eta_B \rangle C_B}{1 - j_0} > 0.
$$

Then, the stability of this state with respect to a small perturbation $\{\delta r_0, \delta r_A, \delta r_B\}$ can be studied by linearizing (15) around the stationary solution, at fixed $\psi_A = \psi_B = 0$. The resulting Jacobian matrix is

$$
\begin{pmatrix}
-1 + j_0 & 0 & 0 \\
0 & -1 + j_s^A F_{AA} + j_a F_{AB} & j_s^B F_{AB} \\
0 & j_s^A F_{AB} + j_a F_{BB} & -1 + j_s^B F_{BB}
\end{pmatrix}
$$

where

$$
\begin{aligned}
F_{AA} &= \int dx \, \rho(x) \, \eta_A^2 \cos(\theta_A)^2 = \tfrac{1}{2} \langle \eta_A^2 \rangle \\
F_{BB} &= \int dx \, \rho(x) \, \eta_B^2 \cos(\theta_B)^2 = \tfrac{1}{2} \langle \eta_B^2 \rangle \\
F_{AB} &= \int dx \, \rho(x) \, \eta_A \eta_B \cos(\theta_A) \cos(\theta_B) = \tfrac{x}{4} \langle \eta_A \rangle \langle \eta_B \rangle.
\end{aligned}
$$

The homogeneous solution (22) is stable if $j_0 < 1$ and if the couplings parameters $\{j_s^A, j_s^B, j_a\}$ satisfy the following system of inequalities:

$$
\begin{cases}
j_a \dfrac{x}{4} \langle \eta_A \rangle \langle \eta_B \rangle + j_s^A \dfrac{1}{2} \langle \eta_A^2 \rangle + j_s^B \dfrac{1}{2} \langle \eta_B^2 \rangle < 2 \\[2mm]
\left(1 - j_s^A \dfrac{1}{2} \langle \eta_A^2 \rangle \right) \left(1 - j_s^B \dfrac{1}{2} \langle \eta_B^2 \rangle \right) - \left(j_a + j_s^A j_s^B \dfrac{x}{4} \langle \eta_A \rangle \langle \eta_B \rangle \right) \dfrac{x}{4} \langle \eta_A \rangle \langle \eta_B \rangle > 0.
\end{cases}
\tag{23}
$$

If $j_0 \geq 1$, the system undergoes an amplitude instability. For values of $\{j_s^A, j_s^B, j_a\}$ that exceed the threshold implicitly defined by (23), there is a region of the four-dimensional space $(\theta_A, \theta_B, \eta_A, \eta_B)$ where the total input $I^{\text{tot}}$ (19) is negative. The dynamics is no longer linear and the activity profile at the fixed point is localized around a particular direction, and characterized by positive stationary values of the order parameters $r_A, r_B$. This 'bump' of activity can be localized either more strongly in map A ($r_A > r_B > 0$), in map B ($r_B > r_A > 0$) or at the same level in both maps ($r_A = r_B > 0$). Since maps A and

B are correlated according to 3, the location of the stationary bump of activity is constrained to be the same in the two maps: $\psi_A = \psi_B \equiv \psi$, with arbitrary $\psi$. The asymmetric term in the connectivity, modulated by the parameter $j_a$, further contributes to aligning the location of the bump in map B to the one in map A. The system can relax to a continuous manifold of fixed points parameterized by $\psi$ that are marginally stable. In this continuous attractor regime, the system can store in memory any direction of motion $\psi$, as the activity is localized in absence of tuned inputs.

Examples of two-dimensional cross-sections of the four-dimensional activity profile in the marginal phase are shown in *Figure 2b*. *Figure 2c* shows one-dimensional cross-sections of the activity as a function of $\theta_A$ or $\theta_B$, for different values of $\eta_A$ and $\eta_B$. These correspond to the tuning curves of the model for stationary external inputs. The tuning profile can have either a full-cosine modulation or a rectified cosine modulation, depending on the values of $\eta_A, \eta_B$. The plots of *Figure 1f-g* correspond to the activity as a function of $\theta_A, \theta_B$ in the case of a discrete number of neurons – each neuron with a different value of $\eta_A, \eta_B$.

## Time-dependent external input

The activity of motor cortical neurons shows no signature of being in a stationary state but instead displays complex transients. To model the data, we assumed that the neurons are responding both to recurrent inputs and to fluctuating external inputs that can be either homogeneous or tuned to $\theta_A, \theta_B$, with peak at constant location $\Phi_A = \Phi_B \equiv \Phi$ (*Equation 6*). The total input 2 that neurons are subject to at a given time $t$ is:

$$I^{\text{tot}}(x; t) = I_0(t) + I_A(t) \, \eta_A \cos(\theta_A - \Phi^{\text{ext}}) + I_B(t) \, \eta_B \cos(\theta_B - \Phi), \tag{24}$$

where

$$
\begin{aligned}
I_0(x; t) = \quad & C_0(t) + \eta_A C_A(t) + \eta_B C_B(t) + j_0 r_0(t) \\
I_A(t) = \quad & j_s^A r_A(t) + \epsilon_A(t) \\
I_B(t) = \quad & j_s^B r_B(t) + j_a r_A(t) + \epsilon_B(t).
\end{aligned}
\tag{25}
$$

In order to draw a correspondence between the model and the data, we note that the activity of each neuron in the data corresponds to the activity rate at a specific coordinate $\theta_A, \theta_B, \eta_A, \eta_B$ in the model. From *Equations 1 and 24* we see that, if we assume that changes in the external inputs happen on a time scale larger than $\tau$, the modulation of the activity profile in map A at each time $t$ is $\cos(\theta_A - \Phi)$ and has amplitude proportional to $\eta_A$, while the modulation of the activity profile in map B is $\cos(\theta_B - \Phi)$ and has amplitude proportional to $\eta_B$. Accordingly, for each recorded neuron $\theta_A$ and $\theta_B$ represent the location of the peak of the neuron's tuning curve computed during the preparatory and movement-related epoch, respectively, while $\eta_A$ and $\eta_B$ are proportional to the amplitude of the tuning curves.

## Fitting the model to the data

Neurons activity rates were computed by smoothing the spike trains with a Gaussian kernel with s.d. of 25ms and averaging them across all trials with the same condition; $r_k^{(i)}(t)$ denotes the rate of neuron $i$ at time $t$ for condition $k$, each condition corresponding to one of the 8 angular locations of the target on the screen. Since trials had highly variable length, we normalized the responses along the temporal dimension before averaging them over trials, as follows. We divided the activity into three temporal intervals: from the target onset to the go cue; from the go cue to the start of the movement; from the start of the movement to the end of the movement. For each interval, we normalized the response times to the average length of the interval across trials. We then aggregated the three intervals together. We defined the preparatory and execution epochs – denoted by A and B – as two 300ms time intervals beginning, respectively, 100ms after target onset and 50ms before the start of the movement, in line with (*Elsayed et al., 2016*). *Figure 3—figure supplement 1* shows that our results do not change qualitatively when the lengths of the preparatory and execution intervals are increased. For each neuron $i$, we fitted the activity rate averaged across time within each epoch as a function of the angular position $\Phi$ of the target with a cosine function:

$$a_\nu^{(i)} + b_\nu^{(i)} \cos(\theta_\nu^{(i)} - \Phi), \quad \nu = A, B,$$

where the parameters $\theta_A^{(i)}$ and $\theta_B^{(i)}$ represent the neuron's preferred direction during the preparatory and execution epochs. In our the model, the direction modulation of the rates, see (25), is proportional to $\eta_{A\backslash B}$, that measures how strongly the neuron participates in the two epochs of movement; hence, we defined $\eta_{A\backslash B}^{(i)}$ to be proportional to the amplitude of the tuning curve:

$$\eta_\nu^{(i)} = b_\nu^{(i)} / \max_i(b_\nu^{(i)}), \quad \nu = A, B. \tag{26}$$

The scatter plot of $\eta_A, \eta_B$ (**Figure 1e**) shows an outlier with $\eta_B \sim 1$. We checked that our results hold true if we discard that point. **Figure 1—figure supplement 2** shows that both $\eta_A$ and $\eta_B$ strongly correlate with the $R^2$-coefficient of the cosine fit. In other words, neurons with a higher value of $\eta_A, \eta_B$ are the ones whose tuning curves more strongly resemble a cosine (this holds true in our simulations too, see **Figure 5—figure supplement 2c**, and Discussion). The order parameters (14) are computed at each time $t$ by approximating the integrals with the sums:

$$
\begin{aligned}
r_0^{\text{data}}(t) &= \frac{1}{N}\frac{1}{n_c}\sum_i\sum_k r_k^{(i)}(t), \\
r_{A/B}^{\text{data}}(t) &= \frac{1}{N}\frac{1}{n_c}\sum_i\sum_k \eta_{A/B}^{(i)}\cos\left(\theta_{A/B}^{(i)} - \psi_{A/B}^{(k)}(t)\right) r_k^{(i)}(t) \quad \text{for} \quad k = 1, \ldots, n_c,
\end{aligned}
\tag{27}
$$

where $N$ is the number of neurons and $n_c = 8$ is the number of conditions. The angular location of the localized activity $\psi_{A/B}^{(k)}(t)$ can be computed from (14) as

$$0 = \frac{1}{N}\sum_i \eta_{A/B}^{(i)}\sin\left(\theta_{A/B}^{(i)} - \psi_{A/B}^{(k)}(t)\right) r_k^{(i)}(t). \tag{28}$$

However, this estimate is strongly affected by the heterogeneity in the distribution of $\theta_A, \theta_B$ - deviating from the rotational symmetry of the model. That is why in **Figure 2** we approximated $\psi_{A/B}^{(k)}(t)$ by the $k - th$ angular location of the target on the screen. We checked that computing $\psi_{A/B}^{(k)}(t)$ by using either method does not affect the dynamics of the order parameters $r_{A/B}^{\text{data}}(t)$.

We assumed that the observed dynamics of the order parameters $r_0(r), r_A(t), r_B(t)$ obeys **Equations 15; 16** with time-dependent external inputs of the form (6). We inferred the value of the parameters $\{C_0(t), C_A(t), C_B(t), \epsilon_A(t), \epsilon_B(t)\}_t$ of the external currents and $j_0, j_s^A, j_s^B, j_a$ of the coupling matrix that allow us to reconstruct the dynamics of the order parameters $r_0, r_A, r_B$ computed from data; since this inference problem is undetermined, we required as further constraint that the model reconstruct the dynamics of the following two additional order parameters:

$$
\begin{aligned}
\tau\frac{d}{dt}r_{0A}(t) &= -r_A(t) + \int\int dx\,\rho(x)\,\eta_A\,[I^{\text{tot}}(\theta_A, \theta_B, \eta_A, \eta_B; t)]_+ \\
\tau\frac{d}{dt}r_{0B}(t) &= -r_B(t) + \int\int dx\,\rho(x)\,\eta_B\,[I^{\text{tot}}(\theta_A, \theta_B, \eta_A, \eta_B; t)]_+.
\end{aligned}
\tag{29}
$$

In this way, for given coupling parameters $j_0, j_s^A, j_s^B, j_a$, we can uniquely identify the external currents parameters that produced the observed dynamics. Still, an equally good reconstruction of $r_0, r_A, r_B, r_{0A}, r_{0B}$ can be obtained for different choices of coupling parameters (2). Hence, we inferred the model parameters by minimizing a cost function composed of two terms: one that is proportional to the reconstruction error of the temporal evolution of the order parameters and the other that represents an energetic cost penalizing large external inputs.

## Fitting the model to the data: details
The fitting procedure was divided in the following steps:

1. The time interval $T$ going from the target onset till the end of the movement was binned into $\Delta t = 5ms$ time bins: $T = \{\Delta t_1, \Delta t_2, \ldots \Delta t_T\}$.
2. The couplings parameters were initialized to zero: $j_0 = 0, j_s^A = 0, j_s^B = 0, j_a = 0$. At the first time bin, the external currents parameters were initialized to zero:
   $C_0(\Delta t_1) = C_A(\Delta t_1) = C_B(\Delta t_1) = \epsilon_A(\Delta t_1) = \epsilon_B(\Delta t_1) = 0$

   and the reconstructed order parameters ($r_0$, …) were initialized to the order parameters estimated from the data ($r_0^{\text{data}}, \ldots$):

$$r_0(\Delta t_1) = r_0^{\text{data}}(\Delta t_1), \quad r_A(\Delta t_1) = r_A^{\text{data}}(\Delta t_1), \quad r_B(\Delta t_1) = r_B^{\text{data}}(\Delta t_1),$$

$$r_{0A}(\Delta t_1) = r_{0A}^{\text{data}}(\Delta t_1), \quad r_{0B}(\Delta t_1) = r_{0B}^{\text{data}}(\Delta t_1).$$

3. For each time step $\Delta t_i, \; i = 2, \dots, T$:
   - We started from the reconstructed order parameters at the previous time step:

   $$r_0(\Delta t_{i-1}), r_A(\Delta t_{i-1}), r_B(\Delta t_{i-1}), r_{0A}(\Delta t_{i-1}), r_{0A}(\Delta t_{i-1}),$$

   and we let the dynamical system (15, 29) with external currents parameters $C_0, C_A, C_B, \epsilon_A, \epsilon_B$ evolve for $\Delta t = 5ms$ to estimate the order parameters at the current time step:

   $$r_0(\Delta t_i), r_A(\Delta t_i), r_B(\Delta t_i), r_{0A}(\Delta t_i), r_{0A}(\Delta t_i).$$

   - We inferred the value of the external currents parameters

   by minimizing the reconstruction error:

   $$E_{\Delta t}(C_0, C_A, C_B, \epsilon_A, \epsilon_B) = \frac{[r_0(\Delta t) - r_0^{\text{data}}(\Delta t)]^2}{\frac{1}{T}\sum_i r_0^{\text{data}}(\Delta t_i)} + \frac{[r_A(\Delta t) - r_A^{\text{data}}(\Delta t)]^2}{\frac{1}{T}\sum_i r_A^{\text{data}}(\Delta t_i)}$$
   $$+ \frac{[r_B(\Delta t) - r_B^{\text{data}}(\Delta t)]^2}{\frac{1}{T}\sum_i r_B^{\text{data}}(\Delta t_i)} + \frac{[r_{0A}(\Delta t) - r_{0A}^{\text{data}}(\Delta t)]^2}{\frac{1}{T}\sum_i r_{0A}^{\text{data}}(\Delta t_i)} + \frac{[r_{0B}(\Delta t) - r_{0B}^{\text{data}}(\Delta t)]^2}{\frac{1}{T}\sum_i r_{0B}^{\text{data}}(\Delta t_i)}, \tag{30}$$

   that quantifies the difference between the order parameters estimated from the data and the reconstructed ones; note that the dependence of the cost function $E$ on $C_0, C_A, C_B, \epsilon_A, \epsilon_B$ is implicitly contained in the reconstructed order parameters. We minimized the cost function (30) by using an interior point method algorithm [**Byrd et al., 1999**] starting from the initial condition

   $$C_0(\Delta t_{i-1}), C_A(\Delta t_{i-1}), C_B(\Delta t_{i-1}), \epsilon_A(\Delta t_{i-1}), \epsilon_B(\Delta t_{i-1});$$

   we imposed that $\epsilon_A > 0, \epsilon_B > 0$ and added a $L1$ regularization term to the external inputs, not to infer pathologically large positive and negative inputs that balance each other. This favors solutions with smaller external inputs and non-zero reconstruction error with respect to ones with larger inputs and almost-zero reconstruction error.
   The external currents inferred with step 3 depend on our initial choice of of the couplings parameters $j_0, j_s^A, j_s^B, j_a$.

4. Using step 3, the value of the couplings parameters $j_0, j_s^A, j_s^B, j_a$ is inferred by minimizing the cost function
   $$E_{\text{tot}} = \alpha E_{\text{rec}} + E_{\text{ext}}$$
   composed of two terms: the reconstruction error and a term that favors small external currents:
   $$E_{\text{rec}} = \frac{1}{T}\sum_{i=2}^{T} \left\{ E_{\Delta t_i} \left[ C_0(\Delta t_i), C_A(\Delta t_i), C_B(\Delta t_i), \epsilon_A(\Delta t_i), \epsilon_B(\Delta t_i) \right] \right\},$$
   $$E_{\text{ext}} = \frac{1}{T}\sum_{i=2}^{T} \left[ |C_0(\Delta t_i)| + |C_A(\Delta t_i)| + |C_B(\Delta t_i)| + \epsilon_A(\Delta t_i) + \epsilon_B(\Delta t_i) \right]. \tag{31}$$
   The minimization is done using a surrogate optimization algorithm [**Gutmann, 2001**].

The result does not depend on the choice of the time bin $\Delta t$. Also, the result weakly depends on the time constant $\tau$ in the mean field equations (e.g. 15), if $\tau$ varies on in the range: $10 - 100ms$. We set $\tau = 25$ ms.

## Simulations of the model

We simulated the dynamics of a finite network of $N$ neurons. To each neuron $i$, we assigned the variables $\theta_A^{(i)}, \theta_B^{(i)}, \eta_A^{(i)}, \eta_B^{(i)}$ as follows.

- In the $\theta_{A\backslash B}$-space, we sampled $N_\theta$ points $\{\theta_A^{(i)}, \theta_B^{(i)}\}_{i=1}^{N_\theta}$ equally spaced along the lines

$$\theta_A - \theta_B = \text{const}$$

in such a way that their joint distribution matches the distribution of the data (4), as shown in *Figure 5—figure supplement 2a*.

- In the $\eta_{A\backslash B}$-space, we drew $N_\eta$ points $\{\eta_A^{(i)}, \eta_B^{(i)}\}_{i=1}^{N_\eta}$ at random from the empirical distribution $\rho_{pA}(\eta_A)\rho_{pB}(\eta_B)$.
- For $i = 1, 2, \dots N_\theta$, we assigned to a block of $N_\eta$ neurons the same coordinates $\theta_A^{(i)}, \theta_B^{(i)}$ in the $\theta_{A\backslash B}$-space, and all possible coordinates $\{\eta_A^{(j)}, \eta_B^{(j)}\}_{j=1}^{N_\eta}$ in the $\eta_{A\backslash B}$-space, so that the overall number of neurons is $N = N_\theta N_\eta$.

The network dynamics we simulated is defined by the following stochastic differential equation:

$$\tau \frac{dr_i(t)}{dt} = -r_i(t) + \left[ \frac{1}{N} \sum_{j=1}^{N} J_{ij} r_j(t) + I_i^{\text{ext}}(t) + \xi_i(t) \right]_+ , \tag{32}$$

$$d\xi_i(t) = -\gamma \xi_i(t) dt + \sigma_n dW, \tag{33}$$

where

$$
\begin{aligned}
J_{ij} &= j_0 + \sum_{\nu=A,B} j_s^\nu \eta_\nu^{(i)} \eta_\nu^{(j)} \cos(\theta_\nu^{(i)} - \theta_\nu^{(j)}) + j_a \eta_B^{(i)} \eta_A^{(j)} \cos(\theta_B^{(i)} - \theta_A^{(j)}), \\
I_i^{\text{ext}} &= C_0(t) + \eta_A^{(i)} C_A(t) + \eta_A^{(i)} \cos(\theta_A^{(i)} - \Phi) \epsilon_A(t) \\
&\quad + \eta_B^{(i)} C_B(t) + \eta_B^{(i)} \cos(\theta_B^{(i)} - \Phi) \epsilon_B(t);
\end{aligned} \tag{34}
$$

$W$ in (33) is a Wiener process, and the parameters $\gamma = 75/s$, $\sigma_n = 0.35 Hz/\sqrt{s}$ set the magnitude of the noise fluctuations. The level of noise is chosen so that the results of the PCA analysis (see next section) match the data. Note that by setting the noise to zero and taking the limit $N \to \infty$ we recover the mean-field *Equation 1*. The results of *Figure 3* are obtained from a network of $N = 16000$ neurons; we simulate the network dynamics for 8 location of the external input $\Phi$ and 10 trials for each of the 8 conditions, that is 10 instances of the noisy dynamics. The order parameters shown in *Figure 3* are computed from single-trial activity, and then averaged over trials. The correlation-based analysis, instead, is obtained from trial-averaged activity.

## Correlation-based analysis

The correlation-based analysis explained in this section was performed both on the smoothed and trial-averaged spike trains from recordings, and on the trial-averaged activity rates from simulations. Out of the 16,000 units in the simulated network model, we considered a subset of the same size $N = 141$ as the recorded neurons. In particular, for each neuron in the data, we chose the corresponding one from simulations with the closest value of parameters $\theta_A, \theta_B, \eta_A, \eta_B$ - the one with smallest euclidean distance in the space defined by $(\theta_A, \theta_B, \eta_A, \eta_B)$. To compute signal correlations, we preprocessed the data as follows, both for the recordings and for the simulations. For each neuron, we normalized the activity by its standard deviation (computed across all times and all conditions); then, we mean-centered the activity across conditions. The $T = 300$ ms long preparatory activity for all $C = 8$ conditions was concatenated into a $N \times TC$ matrix denoted by $P$, and the movement-related activity was grouped into an analogous matrix $M$.

CCA: We used a canonical correlation analysis (CCA) to compare the population activity from data and simulations. First, we projected the activity onto the $K$ PCA dimensions that captured 90% of the activity variance across conditions; for the preparatory epoch, $K = 20$ for both the data and the simulations; for the movement-related epoch, $K = 14$ for the data, and $K = 10$ for the simulations. We then applied CCA to look for common patterns in the activity matrices from data and simulations. In brief, CCA finds a set of linear transformations of the activity matrices, in such a way that the transformed variables (called canonical variables) are maximally correlated. *Figure 5a* shows a high correlation between the sets of canonical variables from data and simulations. We repeated the procedure for the simulations of the purely feedforward network. In this case, the movement-related activity is slightly higher dimensional than for the recurrent network ($K = 12$), and the average canonical correlation is a bit lower. Note that both activities are trial-averaged, and the same realization of the noise was used in both sets of simulations (feedforward and recurrent).

PCA: We obtained correlation matrices relative to preparatory and movement related activity by computing the correlations between the rows of the respective matrices. We then identified the prep-PCs and move-PCs by performing PCA separately on the matrices $P$ and $M$. The degree of orthogonality between the prep- and move- subspaces was quantified by the Alignment Index $A$ (*Elsayed et al., 2016*), measuring the amount of variance of the preparatory activity explained by the first $K$ move-PCs:

$$A = \frac{\text{Tr}(E_{\text{mov}}^T C_{\text{prep}} E_{\text{mov}})}{\sum_{i=1}^{K} \sigma_{\text{prep}}(i)},$$

where $E_{\mathrm{mov}}$ is the matrix defined by the top $K$ move-PCs, $C_{\mathrm{prep}}$ is the covariance matrix of the preparatory activity and $\sigma_{\mathrm{prep}}(i)$ is the $i$-th eigenvalue of $C_{\mathrm{prep}}$ was set to the number of principal components needed to explain 90% of the execution activity variance. Hence, the Alignment Index ranges from 0 (orthogonal subspaces) to 1 (aligned subspaces). As random test, we computed the Random Alignment Index between two sets of $K$ dimensions drawn at random within the space occupied by neural activity, using the Monte Carlo procedure described in *Elsayed et al., 2016*. We performed the same analysis on both the data ($K = 14$) and the model ($K = 10$) trial averaged activity.

The dimensionality of the preparatory and movement-related subspaces can be understood as follows. The activity of the ring model encoding the value of a singular angular variable is two-dimensional in the Euclidean space. Similarly, the activity of the double-ring model encoding two distinct circular maps is four-dimensional. Our model is an extension of the double-ring model, where both the connectivity matrix and the external fields (*Equations 5; 6*) are a sum of several terms, each one composed of an $\eta$-dependent term multiplying a $\theta$-dependent term. The connectivity matrix is still rank-four, but its eigenvectors are modulated by the $\eta$ variables. Although the dynamics is four-dimensional, we have shown that during movement preparation the activity is localized only in map A, while during movement execution it is predominantly localized in map B: in either epoch, we expect only two eigenvalues to explain most of the activity variance, as we indeed see from simulations in absence of additive noise (*Figure 6—figure supplement 1*). The noise term introduces extra random dimensions; as the dynamics gets higher dimensional, both the alignment index and the random alignment index get smaller.

jPCA: We quantified the rotational structure in the data by applying the jPCA (*Churchland et al., 2012*) dimensionality reduction technique to both simulated activity and recordings during movement execution. jPCA is a technique to identify the dimensions that capture rotational dynamics in the data. Given the matrix of activity $M$ defined above, jPCA finds the best fit for the real skewed-symmetric matrix $R$ that transforms the activity $M$ into its derivative $\dot{M}$:

$$\dot{M} = RM.$$

The plane capturing the strongest rotations is spanned by the eigenvectors of the matrix $R$ associated with the two largest eigenvalues. *Figure 5—figure supplement 4* shows that the trajectories from simulations qualitatively resembled the ones from data when projected onto the jPCA subspace that capture most of the variance.

## EMG signals

The EMG signals were recorded from macaque monkey Bx during the execution of a center-out reaching movement, as described in *Balasubramanian et al., 2020*. Bi-polar electromyographic electrodes were implanted in 13 individual muscles (Anterior Deltoid, Posterior Deltoid, Pectoralis Major, Biceps Lateral, Biceps Medial, Triceps Long head, Triceps Lateral, Brachioradialis, Flexor Carpi Ulnaris, Flexor Digitorum Superficialis, Extensor Carpi Radialis, Extensor Digitorum Communis, Extensor Carpi Ulnaris). The signals were amplified individually and bandpass filtered between 0.3 and 1 kHz prior to digitization and sampled at 10 kHz. The EMG signals shown in *Figure 5b* were rectified and smoothed; signals from different trials were rescaled to match the average length of movement execution, and then averaged over trials. The linear readout of the activity from simulations was defined as

$$z_k(t) = \sum_{j=1}^{N} W_{k,j} r_j(t) + W_{k,0} \tag{35}$$

where $r_j$ is the trial-averaged activity from simulations, with the $C = 8$ conditions concatenated into a matrix of dimensions $N \times TC$, where $T$ the average duration of movement execution; $z$ is a matrix with dimension $m \times TC$, $m = 13$ being the number of recorded muscles. For each muscle $k$, the EMG signal $z_k(t)$ was regressed onto the activity rate $\{r_j(t)\}_j$ to find the readout weight vector $\{W_{k,j}\}_{j=0}^{N}$. For the reconstructed EMG signals shown in *Figure 5b*, we used as inputs the activity of N = 1000 units drawn at random from the 16000 units in the network model. For each muscle $k$, we cross-validated the regression by randomly partitioning the TC time-steps into 10 folds of non-consecutive time points. The decoder was trained on 9 folds using LASSO regression, and tested on the left-out fold. We repeated the procedure 10 times and reported a normalized mean squared error of 0.0066.

## Acknowledgements

We thank Wei Liang from the Hatsopoulos lab for providing the EMG signals; Jason MacLean, Alex P Vaz, Subhadra Mokashe, Alessandro Sanzeni for helpful discussions; and Stephen H Scott for pointing the work of *Nashef et al., 2021* to our attention. This work has been supported by NIH R01NS104898.

## Additional information

### Funding

| Funder | Grant reference number | Author |
|---|---|---|
| National Institutes of Health | R01NS104898 | Nicolas Brunel |

The funders had no role in study design, data collection and interpretation, or the decision to submit the work for publication.

### Author contributions

Ludovica Bachschmid-Romano, Conceptualization, Software, Formal analysis, Investigation, Writing - original draft; Nicholas G Hatsopoulos, Resources, Data curation, Funding acquisition, Writing – review and editing; Nicolas Brunel, Conceptualization, Resources, Supervision, Funding acquisition, Investigation, Writing – review and editing

### Author ORCIDs

Ludovica Bachschmid-Romano http://orcid.org/0000-0002-7249-5167
Nicolas Brunel http://orcid.org/0000-0002-2272-3248

### Ethics

The data analyzed in this paper has been previously published in Rubino, Robbins, Hatsopoulos, Nature neuroscience. 2006 Dec;9(12):1549-57, where the ethics approval is described. No ethical approval was necessary for this article because no new experiments were performed.

### Decision letter and Author response

Decision letter https://doi.org/10.7554/eLife.77690.sa1
Author response https://doi.org/10.7554/eLife.77690.sa2

## Additional files

### Supplementary files

• Transparent reporting form

### Data availability

Source data and all the codes used for data analysis will be made publicly available at https://github.com/lbachromano/M1_Preparatory_Movement_Representation (copy archived at *Bachschmid-Romano et al., 2023*).

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
