## [Editor Report]

The study develops a recurrent network model of M1 for center-out reaches, starting from a conventional tuning (or representational) perspective. Through recurrent connectivity, the model shows uncorrelated tuning for movement direction during preparation and execution with the dynamic transition between the two states. The continuous attractor model provides an important example of flexible switching between neural representations and is supported by convincing simulations and analysis.

---

## [Decision Letter]

**Decision letter after peer review:**

Thank you for submitting your article "Interplay between external inputs and recurrent dynamics during movement preparation and execution in a network model of motor cortex" for consideration by *eLife*. Your article has been reviewed by 3 peer reviewers, and the evaluation has been overseen by a Reviewing Editor and Ronald Calabrese as the Senior Editor. The following individual involved in the review of your submission has agreed to reveal their identity: Guillaume Hennequin (Reviewer #1).

Essential revisions:

You will find the extensive comments from the reviewers attached to this decision. When you submit your revision, please provide a response to each of these in turn. The most important 3 areas that require your special attention are:

a) Please justify the choice of the hyperparameters (especially \α) in the model and/or explore the sensitivity of the conclusion to that parameter. (i.e see reviewer 2, comment 5)

b) The "representational" approach taken in this paper should be more clearly contrasted with the current "dynamical" approach – more importantly, however, a fuller evaluation of the model prediction on neural data is required (see reviewer 3).

c) Overall, the clarity of the paper should be improved – the extensive comments below should hopefully provide some indication of where changes are needed.

*Reviewer #1 (Recommendations for the authors):*

I am sorry to have to say that the writing was really difficult to follow. Some members of my group read the paper on biorxiv and gave up halfway through. For me, it was very difficult to follow in the first pass, a bit better in the second pass, and it only "clicked" in the third pass. In general, there is not enough high-level narrative/heads up about where the story is headed. The order in which things are presented appears a bit odd at times. Much of this could be fixed by asking naive friends to lend a critical eye on clarity?

Perhaps the worst section for me was the one on "Time-dependent activity profiles"; on first reading, it was hard to know where the authors are going with this reduction of recurrent inputs to effective local input"; why do we need this? In Eq 6, it becomes hard to know which variables are parameters of the model which the authors control/fit to data, and which ones result from the dynamics of the model. Accordingly, it would be nice to include a recap of the parameters that are optimized in the paragraph starting "the value of the parameters that best fit the data […]" in the next section. Overall this section and the next need streamlining to give a better account of the big picture; and don't you want to start by explaining that the joint density of (thetaA, thetaB, etaA, etaB) is extracted from the data according to Eqs 7, 8 + kernel density estimation? It somehow takes forever to get there.

---

## [Author Response]

Essential revisions:You will find the extensive comments from the reviewers attached to this decision. When you submit your revision, please provide a response to each of these in turn. The most important 3 areas that require your special attention are:a) Please justify the choice of the hyperparameters (especially \α) in the model and/or explore the sensitivity of the conclusion to that parameter. (i.e see reviewer 2, comment 5)b) The "representational" approach taken in this paper should be more clearly contrasted with the current "dynamical" approach – more importantly, however, a fuller evaluation of the model prediction on neural data is required (see reviewer 3).c) Overall, the clarity of the paper should be improved – the extensive comments below should hopefully provide some indication of where changes are needed.

We thank the editor and reviewers for providing constructive feedback. We have revised the manuscript following the reviewers’ suggestions. We substantially rewrote the main sections of the paper and added new plots, mainly to address:

a) The degeneracy of solutions and the choice of the hyperparameter α. Instead of focusing on one solution for a particular value of α, we now thoroughly discuss how different solutions depend on the choice of the model parameters/hyperparameters. We have added new supplementary figures (Figure 4—figure supplement 1 and Figure 4—figure supplement 2) to support our discussion and rewrote the Abstract, Introduction and Discussion accordingly.

b) The comparisons between neuronal activity in the model and in the data. We included three new analyses shown in Figure 5. At the population level, a canonical correlation analysis identifies patterns of activity between simulations and recordings that strongly correlate (Figure 5a); At the output level, a linear redout of neural activity produces patterns of muscle activity that match the ones from recordings (Figure 5.b); At the level of single neurons, a side-by-side comparison between the time course of neuronal activity in the model and in the data shows a good qualitative agreement (Figure 5.c). Finally, we have included a new section in the Discussion, “A dynamical system approach based on tuning to movement direction” to clarify how our work relates to "representational" vs "dynamical" approaches.

To improve clarity, we have significantly rewritten a large part of the paper.

Reviewer #1 (Recommendations for the authors):I am sorry to have to say that the writing was really difficult to follow. Some members of my group read the paper on biorxiv and gave up halfway through. For me, it was very difficult to follow in the first pass, a bit better in the second pass, and it only "clicked" in the third pass. In general, there is not enough high-level narrative/heads up about where the story is headed. The order in which things are presented appears a bit odd at times. Much of this could be fixed by asking naive friends to lend a critical eye on clarity?Perhaps the worst section for me was the one on "Time-dependent activity profiles"; on first reading, it was hard to know where the authors are going with this reduction of recurrent inputs to effective local input"; why do we need this? In Eq 6, it becomes hard to know which variables are parameters of the model which the authors control/fit to data, and which ones result from the dynamics of the model. Accordingly, it would be nice to include a recap of the parameters that are optimized in the paragraph starting "the value of the parameters that best fit the data […]" in the next section. Overall this section and the next need streamlining to give a better account of the big picture; and don't you want to start by explaining that the joint density of (thetaA, thetaB, etaA, etaB) is extracted from the data according to Eqs 7, 8 + kernel density estimation? It somehow takes forever to get there.

Thank you for this feedback. We rewrote most of the paper to better explain the high-level narrative, and followed your suggestions for ‘Time-dependent activity profiles’.